# Disentangled Representation Learning via Modular Compositional Bias

**Whie Jung**      **Dong Hoon Lee**      **Seunghoon Hong**
KAIST
{whieya, donghoonlee, seunghoon.hong}@kaist.ac.kr

## Abstract

Recent disentangled representation learning (DRL) methods heavily rely on factor-specific strategies—either learning objectives for attributes or model architectures for objects—to embed inductive biases. Such divergent approaches result in significant overhead when novel factors of variation do not align with prior assumptions, such as statistical independence or spatial exclusivity, or when multiple factors coexist, as practitioners must redesign architectures or objectives. To address this, we propose a compositional bias, a modular inductive bias decoupled from both objectives and architectures. Our key insight is that different factors obey distinct "recombination rules" in the data distribution: global attributes are mutually exclusive, *e.g.,* a face has one nose, while objects share a common support (any subset of objects can co-exist). We therefore randomly remix latents according to factor-specific rules, *i.e.,* a mixing strategy, and force the encoder to discover whichever factor structure the mixing strategy reflects through two complementary objectives: (i) a prior loss that ensures every remix decodes into a realistic image, and (ii) the compositional consistency loss introduced by Wiedemer et al. [50], which aligns each composite image with its corresponding composite latent. Under this general framework, simply adjusting the mixing strategy enables disentanglement of attributes, objects, and even both, without modifying the objectives or architectures. Extensive experiments demonstrate that our method shows competitive performance in both attribute and object disentanglement, and uniquely achieves joint disentanglement of global style and objects. Code is available at https://github.com/whieya/Compositional-DRL.

## 1 Introduction

Understanding the underlying structure of data has become increasingly important for building robust and interpretable machine learning models. A key approach to addressing this challenge is unsupervised disentangled representation learning (DRL) [1, 16], which aims to factorize data into its fundamental compositional concept representations. By interpreting the world through compositional concepts, it becomes feasible to decompose unseen data into simpler, more interpretable components. Moreover, this approach dramatically improves the data efficiency of learning [24, 26], as unseen data can be explained as combinations of already learned concepts.

As a key theoretical insight in DRL, Locatello et al. [30] show that disentangled representations aligned with underlying ground-truth factors of variation cannot be reliably achieved without appropriate inductive biases or explicit supervision. Consequently, designing a method to embed suitable inductive biases, which impose constraints on the model or assumptions about the data, has become a central challenge in DRL. Specifically, attribute disentanglement methods often enforce information-theoretic objectives [29, 52], while object disentanglement methods leverage spatial exclusivity through architectural components, such as slot attention encoders [19, 31, 41], or additive decoders [2, 25].

39th Conference on Neural Information Processing Systems (NeurIPS 2025).

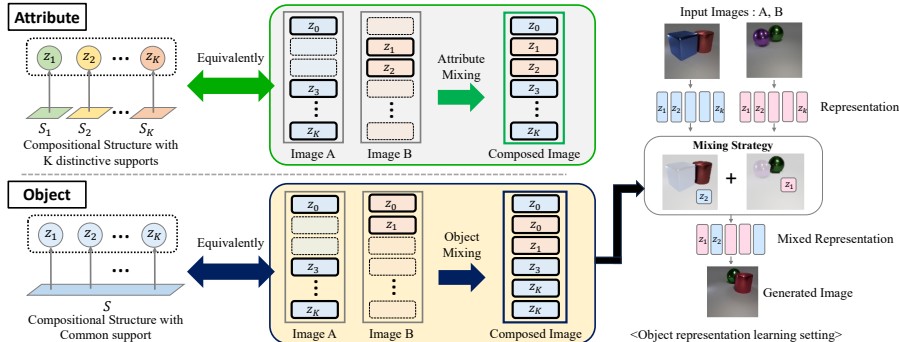

Figure 1: Overview of our method. To derive a compositional bias, we analyze the compositional structure of attribute and object, and implement it as a mixing strategy. Given a mixing strategy, we decode composite representations into an image and minimize our prior loss and consistency loss to ensure it is both realistic and aligned with the latents. Note that the figure illustrates a specific example for object mixing strategy.

Despite impressive progress in each domain, current approaches are constrained by divergent factor-specific strategies for embedding inductive biases, either through learning objectives or architectural components. These vastly different, factor-specific designs embedded within architectures or objective functions require substantial engineering when encountering a novel factor of variation that may violate prior assumptions, such as statistical independence or spatial exclusivity, or when different factors co-occur, as practitioners must redesign architectures or loss functions. Identifying suitable objective functions or architectural components for new factors is challenging, given the vast design space. This motivates the need for a general framework that simplifies inductive biases into a modular component, applicable to various disentanglement factors.

As a first step toward this goal, we propose a *compositional bias*, a modular inductive bias decoupled from both learning objectives and architectures, enabling disentanglement of different factors within a single framework. In particular, we formulate DRL as a process of maximizing compositionality, where the compositional bias characterizes factor-specific biases by defining valid ways to compose disentangled representations. Given two sets of latent representations from different images, we construct a composite representation by exchanging random subsets of latents and maximizing the validity of the resulting composite image, measured by data likelihood and compositional consistency [50]. Analyzing the compositional structures of attributes and objects, we derive specific compositional biases, or *mixing strategies*, that determine valid compositions for attribute and object disentanglement. Incorporating factor-specific biases into this modular mixing strategy, rather than into the architecture or objective function, allows our method to disentangle objects and attributes under a single framework by simply switching the mixing strategy. Our contributions are as follows:

- We propose a disentanglement framework that decouples factor-specific inductive biases from learning objectives and architectures, enabling both attribute and object disentanglement under a single set of objectives and architectures.

- We derive mixing strategies as a compositional bias, embedding factor-specific biases in a modular way and propose specific strategies for attribute, object, and joint disentanglement.

- We compare our methods against baselines specifically designed for attribute or object disentanglement. For attribute disentanglement, our method achieves the best DCI [10], while on object-level tasks, it performs comparably to state-of-the-art methods. Notably, our method demonstrates joint disentanglement of both factors within a single framework.

## 2   Background

In this section, we review the two main streams of disentangled representation learning: attribute and object [1] disentanglement. In particular, we explain how current approaches incorporate factor-specific inductive biases into either learning objectives or model architectures, and why this tight coupling limits the ability to generalize these biases across different factors of variation. For a more comprehensive review of the literature, we refer the reader to Appendix A.3.

---

[1]   We refer to *attribute* factors as properties shared globally across an entire scene, *e.g.*, color, style, while *object* factors are distinct spatial entities within a scene, such as individual objects.

**Attribute disentanglement** In attribute disentanglement [3, 5, 6, 22, 37], scenes are assumed to consist of a fixed set of random variables [22], with methods typically enforcing *statistical independence* among latent dimensions through learning objectives to achieve attribute representations. For example, [3, 5, 22] incorporate Total Correlation [48] into the VAE framework, while [29, 37] use contrastive regularization to ensure that variations in each latent lead to distinct changes in the output space. More recently, [52] proposed minimizing an upper bound on mutual information among latent variables. These *information-theoretic objectives* work well when the data consist of a fixed set of factors, with each latent variable corresponding to a specific factor. However, incorporating such biases directly into the learning objective is not straightforward in general. For instance, in object-centric scenes with varying numbers of objects and permutation invariance, defining information-theoretic objective becomes non-trivial. As a result, practitioners must devise entirely new objectives or models for novel scenarios, such as object-centric scenes. This underscores how embedding the bias directly into the objective function limits generalization to new factors of variation.

**Object disentanglement** Object-centric learning [4, 11, 14, 31] typically models a scene as an unordered set of object representations sampled from a shared generative process, such that permuting these representations does not affect the rendered image. Since measuring independence among object representations is challenging, existing methods often rely on architectural biases that enforce *spatial exclusivity*. Early approaches implement this by rendering each latent as a pair of an image and a mask, then blending these pairs to form the final output [4, 11, 14, 28], with each mask corresponding to a distinct region. Slot attention-based methods [19, 31, 41] similarly impose spatial exclusivity in the encoder, assigning each latent to specific spatial locations in the input. Although these biases effectively disentangle objects, they fail to capture global attributes, such as global style, which are defined for the entire image, as their underlying assumptions no longer hold. Consequently, whenever these assumptions are violated, the architecture must be redesigned, which is non-trivial and costly.

## 3 Method

Our goal is to design a modular inductive bias for DRL, decoupled from learning objectives and architectures. In this section, we present overall formulation to maintain identical learning objectives and architectures across different factors (Sec. 3.1), how we derive a modular inductive bias for different types of factors (Sec. 3.2), and detail specific learning objectives of our framework (Sec. 3.3).

### 3.1 DRL via Maximizing Compositionality

To modularize inductive bias for disentanglement, we construct a framework where learning objective and model architecture are not factor-specific. We formulate the learning objective of DRL as maximizing the *validity* of composite images generated by random recombination of latent representations, where *validity* is measured by data likelihood and consistency with given composite representations. This objective enforces recombination of disentangled factors to yield realistic images, without introducing any factor-specific biases into the model architecture or objective function.

Let $\mathbf{x} \in \mathbb{R}^{H \times W \times C}$ be an input image, and $\mathbf{z} = \{\mathbf{z}_i\}_{i=1}^K$ be a corresponding set of $K$ latent representations, where each $\mathbf{z}_i \in \mathbb{R}^D$ capturing independent factors of variation. Building on an autoencoding framework, the encoder $E_\theta$ maps $\mathbf{x}$ into $\mathbf{z}$ and the decoder $D_\phi$ reconstructs $\mathbf{x}$ from $\mathbf{z}$. To further regularize the compositionality of representations, we first generate composite images $\mathbf{x}^c$ by decoding concatenated random subsets of latents from two images $\mathbf{x}^1, \mathbf{x}^2$. Formally, we define $\mathbf{x}^c = D_\phi(\pi(E_\theta(\mathbf{x}^1), E_\theta(\mathbf{x}^2)))$, where $\pi(\cdot, \cdot) : \mathbb{R}^K \times \mathbb{R}^K \to \mathbb{R}^K$ is a stochastic mixing operator between two sets of latents. We refer to specific designs of $\pi(\cdot, \cdot)$ for disentangling different factor types as the *mixing strategy*, which will be detailed in Sec. 3.2. Given $\mathbf{x}^c$, the compositionality of the representations is maximized through the prior loss and compositional consistency loss ( Sec. 3.3). Note that factor-specific biases will be embedded in *mixing strategy*, while the overall architectures and learning objectives remain unchanged regardless of the factors.

While Jung et al. [21] explored a similar compositional objective, their work focused primarily on object disentanglement and relied on object-specific architectures, e.g., a slot-attention encoder, thereby limiting its applicability to other factors such as attributes. In contrast, our approach imposes no factor-specific bias on either the architectures or the objectives. Instead, we focus on demonstrating

how factor-specific biases can be injected via mixing strategy, allowing disentanglement of different types of factors within a single framework.

## 3.2 Mixing Strategy

In this section, we illustrate how factor-specific inductive biases can be implemented with mixing strategies. We first note that not all compositions of individual factors from different images produce valid images. For instance, when recombining the ground-truth (GT) factors of two face images, compositions containing two noses are invalid. This is because GT factors follow a particular *compositional structure* to form a complete image. Guided by this compositional structure, we formalize valid recombinations of factors using the factorized support assumption [39]. It assumes that the support of disentangled factors' distribution factorizes over individual factors. Formally, let us denote the support of $p(\mathbf{z})$ as $\mathcal{S}(p(\mathbf{z})) = \{\mathbf{z}|p(\mathbf{z}) > 0\}$. Then $\mathbf{z}$ has a factorized support if $\mathcal{S}(p(\mathbf{z})) = \mathcal{S}(p(\mathbf{z}_1)) \times \mathcal{S}(p(\mathbf{z}_2)) \times \cdots \times \mathcal{S}(p(\mathbf{z}_K))$, where $\times$ denotes the Cartesian product. It implies that any random combination of $\mathbf{z}_i$ extracted from distinct images corresponds to some valid image $\mathbf{x}$. Based on this assumption, we analyze the different properties of each factor's support and derive mixing strategies between two images [2] that define valid compositions.

**Mixing Strategy for Attribute Disentanglement**  In attribute disentanglement, it is typically assumed that each scene consists of $K$ unique factors [22, 52]. For example, a ball consists of a fixed set of features such as color, texture, material, etc, with each factor being distinct and appearing only once. This implies that each latent $\mathbf{z}_i$ has a distinct support $\mathcal{S}(p(\mathbf{z}_i)) \neq \mathcal{S}(p(\mathbf{z}_j))$ for all $i \neq j$. Therefore, when combining each latent $\mathbf{z}_i$ from different images into a composite representation $\mathbf{z}^c$, each latent $\mathbf{z}_i$ should be uniquely sampled from its own support. This suggests that the mixing strategy between $\mathbf{z}^1, \mathbf{z}^2$ should guarantee mutual exclusiveness, i.e., each latent $\mathbf{z}_i$ is drawn from one of the two images, but never from both, so that the resulting $\mathbf{z}^c$ always contains $K$ distinct factors. Formally, let $I \subseteq \{1, \ldots, K\}$ be a randomly sampled subset of the index set. The mixing strategy $\pi_{attr}$ for attribute disentanglement is defined as:

$$\pi_{attr}(\mathbf{z}^1, \mathbf{z}^2) = \{\mathbf{z}_j^1 | j \in I\} \cup \{\mathbf{z}_j^2 | j \in I^c\}. \tag{1}$$

**Mixing Strategy for Object Disentanglement**  Object-centric learning often decomposes a scene into $K$ interchangeable representations, each encoding single object instance. Since each object can exist independently of other objects, replacing an object in one image with any object from another image still yields a realistic composition. Formally, this means all $\mathbf{z}_i$ share the same support, *i.e.*, $\mathcal{S}(p(\mathbf{z}_i)) = \mathcal{S}(p(\mathbf{z}_j))$ for $i, j \in \{1, \ldots, K\}$. Consequently, replacing $\mathbf{z}_i$ with any $\mathbf{z}_j$ from different images remains within the valid support of $p(\mathbf{z})$. The mixing strategy for object disentanglement therefore involves randomly sampling $K$ elements from the joint set of $\mathbf{z}^1$ and $\mathbf{z}^2$. Unlike the mixing strategy for attributes, this approach permits unrestricted exchanges between $\mathbf{z}_i^1$ and $\mathbf{z}_j^2$ at different indices (see Figure 1). Specifically, let $I^1, I^2 \subseteq \{1, \ldots, K\}$ be randomly sampled subsets of the index set. Then the corresponding mixing strategy $\pi_{obj}$ is defined as:

$$\pi_{obj}(\mathbf{z}^1, \mathbf{z}^2) = \{\mathbf{z}_j^1 | j \in I^1\} \cup \{\mathbf{z}_j^2 | j \in I^2\}, \text{where } |I^1| + |I^2| = K \tag{2}$$

**Mixing Strategy for Joint Disentanglement**  Unlike prior assumptions that consider either only attributes or objects, real-world scenes often contain both attribute factors and multiple objects, *e.g.*, images with several objects rendered in varying artistic styles. While objective-level biases and model architectures in previous work require non-trivial modifications to capture both types of factors (Sec. 2), our framework naturally extends to this scenario. As attribute factors require mutual exclusiveness while object factors permit arbitrary exchange, we simply partition the $K$ latents into the first $M$ latents for attributes and the remaining $K - M$ latents for objects, and apply the corresponding mixing strategies to each. Formally, let $\mathbf{z}_{1:M}^1$ and $\mathbf{z}_{1:M}^2$ be the $M$ attribute latents, and $\mathbf{z}_{M+1:K}^1$ and $\mathbf{z}_{M+1:K}^2$ the $K - M$ object latents. Then the mixing strategy $\pi_{joint}$ is defined as:

$$\pi_{joint}(\mathbf{z}^1, \mathbf{z}^2) = \pi_{attr}(\mathbf{z}_{1:M}^1, \mathbf{z}_{1:M}^2) \cup \pi_{obj}(\mathbf{z}_{M+1:K}^1, \mathbf{z}_{M+1:K}^2) \tag{3}$$

where $\pi_{attr}$ and $\pi_{obj}$ are defined in Eqs. (1) and (2), respectively.

---

[2]    We demonstrate that mixing from two images is equivalent to mixing from multiple images (Appendix A.4)

### 3.3 Learning Objectives

Given a composite image $\mathbf{x}^c$, we maximize its *validity* measured by likelihood $p(\mathbf{x}^c)$ and consistency with $\mathbf{z}^c$. Note that maximizing the likelihood $p(\mathbf{x}^c)$ alone may lead to degenerate solutions that generate realistic images irrelevant to $\mathbf{z}^c$. The compositional consistency objective prevents it by ensuring the alignment between $\mathbf{x}^c$ and $\mathbf{z}^c$. Below, we provide a detailed description of each objective.

**Maximizing Likelihood of Composite Images**  To maximize the likelihood of $\mathbf{x}^c$, we leverage a pre-trained diffusion model $G_\psi$ for its strong mode coverage [51] and compositional generalization capability [32]. Since denoising loss in diffusion models serves as an upper bound for the negative log-likelihood [17], minimizing the denoising loss with respect to $\mathbf{x}^c$ is one way to increase the likelihood $p(\mathbf{x}^c)$. However, due to the expensive and noisy computation of gradients in back-propagating through a diffusion decoder, we approximate the gradient to optimize $p(\mathbf{x}^c)$ as in [21, 34]:

$$\nabla_\theta \mathcal{L}_{\text{Prior}}(\theta) = \mathbb{E}_{t,\epsilon}[w_t(G_\psi(\mathbf{x}_t^c, t) - \epsilon)\frac{\partial \mathbf{x}^c}{\partial \theta}], \tag{4}$$

where $t \sim \mathcal{U}(t_{\min}, t_{\max})$ is a timestep, $w_t$ is a timestep-dependent function, $\epsilon \sim \mathcal{N}(\mathbf{0}, \mathbf{I})$ is a Gaussian noise. $\mathbf{x}_t^c = \sqrt{\bar{\alpha}_t}\mathbf{x}^c + \sigma_t \epsilon$ denotes a noised image of $\mathbf{x}^c$ and $w_t$ is usually set to $\sigma_t^2$ [34].

We emphasize that we use a pre-trained, frozen, *unconditional* diffusion model for likelihood maximization. This is in contrast to L2C [21] that employs a representation-conditioned diffusion model $G(\mathbf{x}|\mathbf{z})$ for likelihood maximization. In L2C, the conditional score function is learned via a denoising loss conditioned on the encoded latent $\mathbf{z}$ of individual images. In this setup, the diffusion model encounters out-of-distribution (OOD) latents $\mathbf{z}^c$ from mixed representations during likelihood maximization, but there is no guarantee that the score estimation under these OOD conditions $G(\mathbf{x}|\mathbf{z}^c)$ will be valid. Instead, our method leverages a pre-trained unconditional diffusion model to directly optimize likelihood and generate realistic images, while avoiding the OOD conditioning issue in L2C. In Appendix A.9, we further compare our method to L2C's approach by additional experiments.

**Compositional Consistency Loss**  Minimizing the prior loss alone could lead to a degenerate solution, such as generating arbitrary realistic composite images $\mathbf{x}^c$ regardless of the given $\mathbf{z}^c$. To prevent such degeneracy, we adopt compositional consistency loss from Wiedemer et al. [50] to ensure alignment between $\mathbf{z}^c$ and the inverted latent $\hat{\mathbf{z}}^c = E_\theta(D_\phi(\mathbf{z}^c))$ so that $\mathbf{x}^c$ faithfully reflects the contents of $\mathbf{z}^c$. However, empirical observations reveal that minimizing the absolute distance directly is insufficient to prevent misalignment between $\mathbf{x}^c$ and $\mathbf{z}^c$. In practice, we find that the $\mathbf{z}$ from all images tend to cluster closely together in the latent space, when directly minimizing the distance between $\mathbf{z}^c$ and $\hat{\mathbf{z}}^c$. This clustering keeps the distance between $\mathbf{z}^c$ and $\hat{\mathbf{z}}^c$ small even when the composite image $\mathbf{x}^c$ does not faithfully reflect $\mathbf{z}^c$. To address this, we instead minimize the *relative* distance between $\mathbf{z}^c$ and $\hat{\mathbf{z}}^c$, *i.e.*, their distance relative to negative samples, which are latents from other random images. This effectively strengthens the penalty on $\hat{\mathbf{z}}^c$ and $\mathbf{z}^c$ even when their absolute distance is small, ensuring that $\mathbf{z}^c$ must not only match $\hat{\mathbf{z}}^c$ but also remain distinguishable from negative samples. Formally, we define the compositional consistency loss, inspired by the InfoNCE loss [33], as:

$$\mathcal{L}_{\text{Con}}(\theta) = -\log \frac{\exp(d(\hat{\mathbf{z}}^c, \mathbf{z}^c)/\tau)}{\sum_{i \in \{1,\ldots,B\}} \exp(d(\hat{\mathbf{z}}^c, \mathbf{z}^i)/\tau)}, \tag{5}$$

where $\tau$ and $d(\cdot)$ denote temperature and cosine similarity, respectively, and $B$ is a batch size. Note that we should consider the correspondence between $\mathbf{z}^c = \{\mathbf{z}_1^c, \ldots, \mathbf{z}_K^c\}$ and $\hat{\mathbf{z}}^c = \{\hat{\mathbf{z}}_1^c, \ldots, \hat{\mathbf{z}}_K^c\}$ to compute the cosine distance. This can be problematic for object disentanglement, as object-disentangled representations can have permuted orders due to our mixing strategy. In this case, we first apply the Sinkhorn-Knopp algorithm [8] to compute a soft assignment between $\mathbf{z}^c$ and $\hat{\mathbf{z}}^c$, then use the assignment-weighted sum of the distances to compute the loss.

**Overall Objectives**  Following the common practice in DRL [19, 21, 22, 52], our framework is built upon the auto-encoding framework. Specifically, instead of directly reconstructing the image, we minimize a denoising objective using a diffusion decoder, following recent methods [21, 52] as:

$$\mathcal{L}_{\text{Recon}}(\theta, \phi) = \mathbb{E}_{\epsilon,t}\left[w_t \cdot \|D_\phi(\mathbf{x}_t, t, E_\theta(\mathbf{x})) - \epsilon\|^2\right], \tag{6}$$

where $\mathbf{x}_t = \sqrt{\bar{\alpha}_t}\mathbf{x} + \sqrt{1 - \bar{\alpha}_t}$ is a noised image of $\mathbf{x}$ with timestep $t$, $\bar{\alpha}_t = \prod_i^t (1 - \beta_i)$ is a schedule function, and $w_t$ is the weighting parameter. As we use the diffusion decoder $D_\phi$, we use iterative

| Method | Cars3D | | Shapes3D | | MPI3D | |
|---|---|---|---|---|---|---|
| | FactorVAE | DCI | FactorVAE | DCI | FactorVAE | DCI |
| FactorVAE [22] | 0.906 | 0.161 | 0.840 | 0.611 | 0.152 | 0.240 |
| $\beta$-TCVAE [5] | 0.855 | 0.140 | 0.873 | 0.613 | 0.179 | 0.237 |
| InfoGAN-CR [29] | 0.411 | 0.020 | 0.587 | 0.478 | 0.439 | 0.241 |
| LD [47] | 0.852 | 0.216 | 0.805 | 0.380 | 0.391 | 0.196 |
| GS [15] | 0.932 | 0.209 | 0.788 | 0.284 | 0.464 | 0.229 |
| DisCo [37] | 0.855 | 0.271 | 0.877 | 0.708 | 0.371 | 0.292 |
| DisDiff-VQ [52] | **0.976** | 0.232 | 0.902 | 0.723 | 0.617 | 0.337 |
| **Ours** | 0.877 | **0.365** | **0.975** | **0.837** | **0.708** | **0.458** |

Table 1: Quantitative results on attribute disentanglement. Our method achieves state-of-the-art performance in almost all of the datasets, except FactorVAE score in Cars3D. We report mean of 10-repeated runs. Standard deviations are in Appendix A.14.

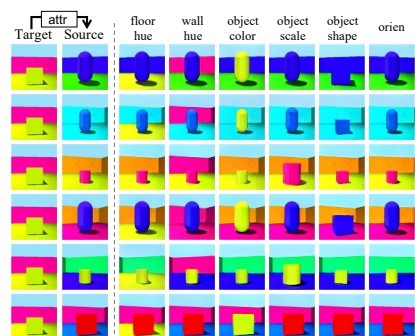

Figure 2: Qualitative results on Shapes3D. Our method identifies all GT factors.

decoding when generating the composite image $\mathbf{x}^c$ from the diffusion decoder, but omit the expression for notational simplicity. The overall objective is given as:

$$\mathcal{L}_{\text{Total}}(\theta, \phi) = \mathcal{L}_{\text{Recon}}(\theta, \phi) + \lambda_{\text{Prior}}\mathcal{L}_{\text{Prior}}(\theta) + \lambda_{\text{Con}}\mathcal{L}_{\text{Con}}(\theta),$$

where $\lambda_{\text{Prior}}$ and $\lambda_{\text{Con}}$ controls the relative importance of the objectives. Note that $\mathcal{L}_{\text{Prior}}$ and $\mathcal{L}_{\text{Con}}$ are optimized only with respect to the encoder parameters ($\theta$), while keeping the decoder parameters ($\phi$) fixed. This prevents unintended cooperation between the encoder and decoder that could generate realistic composite images from suboptimal latent representations.

## 4 Experiment

**Implementation Details**   We implement the decoder $D_\phi$ as a latent diffusion model built on a pre-trained VAE following [21, 52]. Since the diffusion decoder operates on VAE features, we design the image encoder to take these VAE features as input. For object disentanglement, we adopt the same encoder architecture used in recent methods [19, 21], but replace the slot attention module with a lightweight transformer (QFormer [27]) to avoid the inherent inductive biases of slot attention. In attribute disentanglement, we follow the DisDiff [52] encoder design. When generating the composite image $\mathbf{x}^c$ from $\mathbf{z}^c$, we use a few steps (1-4 steps) DDIM [44] sampling to reduce the computational cost of iterative decoding. Because backpropagating gradients through all decoding steps is often prohibitive, we follow recent work on diffusion model fine-tuning [7, 35] to truncate the gradient at the last decoding iteration. For the generative prior $G_\psi$, we train an unconditional latent diffusion model on each training dataset from scratch. See Appendix A.6 for additional implementation details.

**Datasets**   For attribute disentanglement, we evaluate our method on Shapes3D [22], Cars3D [36], MPI3D [13], which are standard datasets in attribute DRL. Each data in these datasets is generated from a fixed number of GT factors. Following [37, 52], all experiments in attribute DRL are conducted at a 64x64 image resolution. For object disentanglement, we use three multi-object datasets, including CLEVR-Easy [42], CLEVR [20], and CLEVR-Tex [42]. Each dataset consists of multiple instances of objects in different object properties, such as colors and shapes. To evaluate joint disentanglement of attributes and objects, we introduce the CLEVR-Style dataset, a new variant of the CLEVR dataset augmented with four distinct artistic styles (see Appendix A.7). A style is regarded as a global attribute of the scene, as it is uniquely determined for each data. This construction yields a complex set of scenes in which both the style and the individual objects must be disentangled. For object- and joint disentanglement, all experiments are conducted at a 128x128 image resolution.

### 4.1 Attribute Disentanglement

We compare our method with: (1) VAE-based methods, including FactorVAE [22] and $\beta$-TCVAE [5], (2) GAN-based methods, including InfoGAN-CR [29], GANspace (GS) [15], LatentDiscovery (LD) [47], and DisCo [37], and (3) the diffusion-based model DisDiff [52]. Note that ours and DisDiff use the same encoder and diffusion-decoder architecture. For methods with vector-wise disentanglement, we follow [9, 52] by applying PCA as a post-processing step for evaluation. We use standard evaluation metrics for disentanglement: the FactorVAE [22] score and the DCI [10] metric. We provide brief descriptions of these metrics in Appendix A.5.

Table 2: Comparisons on object property prediction. Ours achieves performance comparable to state-of-the-art methods. For the position* of CLEVREasy, we use the discrete labels in the dataset and reports the accuracy.

| Method | CLEVREasy | | | CLEVR | | | | CLEVRTex | | |
|---|---|---|---|---|---|---|---|---|---|---|
| | Shape ($\uparrow$) | Color ($\uparrow$) | Position* ($\downarrow$) | Shape($\uparrow$) | Color($\uparrow$) | Material($\uparrow$) | Position($\downarrow$) | Shape($\uparrow$) | Material($\uparrow$) | Position($\downarrow$) |
| SA | 72.25 | 72.33 | 44.08 | 79.4 | 91.30 | 93.18 | 0.064 | 30.44 | 7.890 | 0.482 |
| SLASH | 86.06 | 89.23 | 46.97 | 83.28 | 92.26 | 93.16 | 0.078 | 53.13 | 37.49 | 0.148 |
| LSD | **96.03** | **98.05** | 50.29 | **87.66** | 91.46 | **94.87** | 0.062 | 68.25 | 51.54 | 0.197 |
| L2C | 92.78 | 93.57 | 47.62 | 73.61 | 74.03 | 86.93 | 0.168 | **71.54** | 51.62 | **0.116** |
| **Ours** | 95.81 | 95.38 | **50.72** | 87.04 | **93.93** | 94.81 | **0.032** | 70.90 | 52.2 | 0.133 |

Figure 3: Qualitative results on object-wise manipulation. Objects marked by red arrows are replaced with those marked by green arrows. It demonstrates that our method effectively disentangles individual objects. We also find *empty* latent (depicted with $\phi$), which makes our approach capable of handling varying number of objects.

**Main Results** In Tab. 1, we compare our method to baselines for attribute disentanglement. Our method outperforms all baselines on the Shapes3D and MPI3D by a clear margin, achieving 8% higher FactorVAE score and a 15.7–21.4% higher DCI metric than the second-best methods. On the Cars3D, our method achieves the highest DCI metric. Notably, our method outperforms the state-of-the-art baseline DisDiff [52] by a clear margin on the Shapes3D and MPI3D. This demonstrates the effectiveness of our compositionality maximization approach, which directly enforces support factorization via a mixing strategy. Meanwhile, our method significantly outperforms FactorVAE [22], which similarly employs random mixing of latents but is constrained by a VAE-based total correlation minimization. In contrast, our method is not restricted by specific model architecture such as VAEs, enabling vector-wise disentanglement and an expressive diffusion decoder, which together lead to stronger performance.

**Qualitative Results** In Fig. 2, we analyze the quality of our disentangled representations by swapping latent representations between images. We encode one target image and six source images into $K$ latent representations each. For each $k \in \{1, ..., K\}$, we construct swapped representations by replacing the $k$-th latent of the source images with the $k$-th latent of the target image, then decode these representations into the resulting images. The result shows our method's effectiveness in attribute disentanglement and compositional image generation. On Shapes3D, our approach successfully identifies all six GT factors of variation. Additional results on Cars3D (Appendix A.10) show it captures three independent factors, enabling controlled manipulation of each factor.

## 4.2 Object Disentanglement

We compare our method with state-of-the-art object-centric approaches that use slot attention encoder: SA [31], SLASH [23], LSD [19], and L2C [21]–where LSD and L2C also employ diffusion decoders. We shared the same latent diffusion architecture for all diffusion-based approaches.

**Object property prediction** Following [19, 21], we evaluate the quality of object representations via an object property prediction. In this task, the correspondence between the object representation and its true label is determined through Hungarian matching using object masks. For the baselines,

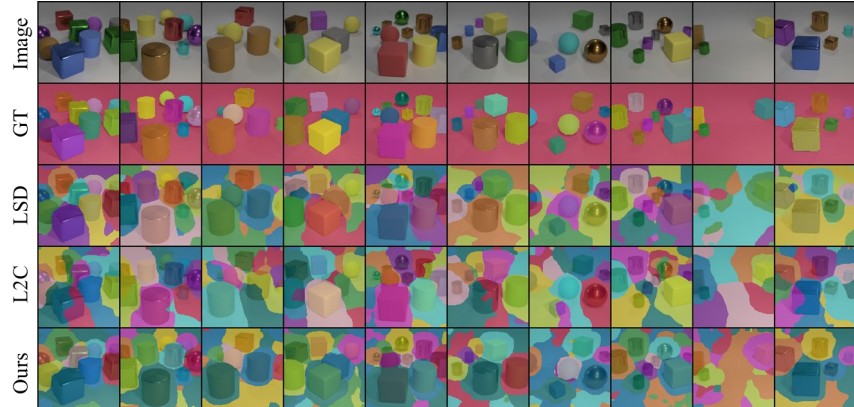

Figure 4: Object segmentation results on CLEVRTex. Despite lacking a built-in spatial clustering mechanism (*e.g.*, slot-attention), our method combined with a Spatial Broadcast Decoder consistently captures complete objects, whereas the baselines often split objects across multiple latents.

slot attention's attention weight produces these masks [19]. Meanwhile, our framework lacks a slot attention module, so we identify each object's region by averaging the difference in output images when composing each representation with others. More details can be found in the Appendix A.8. Tab. 2 demonstrates that our method achieves competitive performance compared to state-of-the-art baselines, LSD, and L2C. Specifically, our method outperforms LSD on CLEVRTex and achieves comparable performance on CLEVR and CLEVR-Easy. It demonstrates the effectiveness of our mixing strategy as an inductive bias for object disentanglement, replacing slot-attention. In comparison to L2C, which also maximizes compositionality, our method performs better on CLEVR and CLEVR-Easy, while being competitive on CLEVRTex. We note that L2C degrades on CLEVR, likely due to undesirable positional biases. Overall, despite our method's broader generality to both attribute and object disentanglement within a single framework, it achieves performance comparable to state-of-the-art methods tailored solely for object disentanglement.

**Object-wise manipulation**   In Fig. 3, we qualitatively explore the compositionality of our object representations. Given pairs of images, we encode each image into $K$ latents, then create a mixed representation by swapping a single latent between images and decode them into composite images. In the figure, we replace one object (red arrow) from the first column with the corresponding object from the first row (green arrow). The result demonstrates that our approach encodes individual objects in a disentangled way, enabling object-wise image manipulation. To be more specific, columns two through five show that each swapped object from the first row is successfully inserted into the first column's image, while the original object is removed from the scene. Moreover, in the fifth row and fifth column, we observe that our method allows the emergence of latent encoding of empty information.

**Unsupervised object segmentation**   We evaluate object representations through unsupervised object segmentation. Unlike slot-attention based methods, which inherently cluster pixels spatially, our approach does not provide a built-in mechanism to group pixels, so object segmentation masks cannot be extracted directly. Nevertheless, to evaluate the quality of object representations, we train a Spatial Broadcast Decoder

Table 3: Comparisons on object segmentation.

| Method | CLEVR | | | CLEVRTex | | |
|---|---|---|---|---|---|---|
| | FG-ARI | mIoU | mBO | FG-ARI | mIoU | mBO |
| LSD | **91.74** | 25.59 | 25.84 | 71.64 | 56.26 | 56.75 |
| L2C | 80.05 | 25.61 | 26.33 | 82.55 | 58.33 | 58.68 |
| Ours | 91.20 | **26.54** | **26.65** | 87.68 | **58.88** | **59.12** |

(SBD) [49] on top of frozen object representations with a reconstruction loss in an unsupervised manner. We followed [14] to extract explicit object masks for each representation using the SBD. We also apply this approach to the slot-attention-based baselines, as it yields better results. We provide more details in the Appendix A.11.

Tab. 3 shows our object segmentation results. On CLEVRTex, our method outperforms both LSD and L2C in FG-ARI, mIoU, and mBO (See Appendix A.5 for definitions of metrics.) On CLEVR, ours achieves the best mIoU and mBO scores, with FG-ARI comparable to LSD. We observe relatively

Table 4: Evaluation of joint disentanglement on CLEVR-Style dataset.

| Method | Style | | Object | | | |
| --- | --- | --- | --- | --- | --- | --- |
| | Acc(↑) | GRAM(↓) | Shape(↑) | Color(↑) | Material(↑) | Position(↓) |
| DisDiff | 0.900 | 13.90 | N/A | N/A | N/A | N/A |
| LSD | 9.500 | 12.97 | 81.64 | 88.93 | 93.42 | 0.076 |
| L2C | 12.20 | 12.86 | **85.02** | **95.35** | **95.35** | **0.046** |
| Ours | **96.50** | **5.050** | 83.56 | 90.48 | 93.74 | 0.053 |

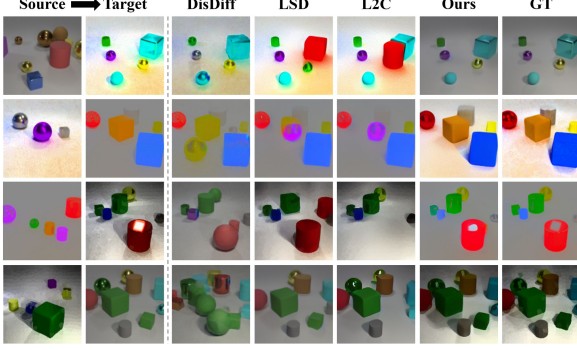

Figure 5: Qualitative comparisons on Style transfer in CLEVR-Style dataset.

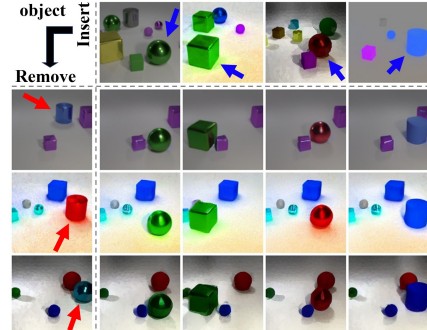

Figure 6: Object manipulation of our method in CLEVR-Style dataset.

lower mIoU and mBO on CLEVR, likely because including the constant background in each object's latent representation does not affect compositional generation, thus avoiding penalties from the compositional loss. Since these backgrounds carry no meaningful information, they do not impact the quality of object representations. Fig. 4 shows qualitative segmentation results on CLEVRTex. Our method consistently encodes complete objects into distinct latents, whereas LSD and L2C often split objects across multiple latents.

## 4.3 Joint Disentanglement of Attribute and Object

In this task, we evaluate the disentanglement of both global attributes (style) and objects in the CLEVR-Style. We compare our method against DisDiff, LSD, and L2C which are state-of-the-art methods in attribute and object disentanglement. For object disentanglement, we evaluate on an object property prediction task. To assess style disentanglement, we sample 1K image pairs with identical content but different styles, swap the style latent of the first image with that of the second, and check whether the synthesized image matches the second. Quantitatively, we report GRAM loss [12] and style prediction accuracy on synthesized images, where we trained a ResNet classifier for style prediction in a supervised manner. While our method can specify the latent for encoding the style, *i.e.*, allocating the first latent as an attribute factor and applying the attribute mixing strategy, the baseline models do not have such mechanisms. To identify the style latent in baselines, we decode all $K \times K$ possible latent exchanges between two images and choose the pair yielding the lowest style loss. To further assess the scalability and robustness of our method on more complex datasets, we also conduct experiments on the *MSN-Style*, an augmentation of the MultiShapeNet (MSN) [45] dataset, which includes over 10k unique, realistic furniture shapes. Further details and results on MSN-Style dataset can be found in Appendix A.7, A.13.

**Main Results** In Tab. 4, our method significantly outperforms all baselines on style disentanglement. Our method achieves over 95% style prediction accuracy, confirming the disentanglement of style information. As information-theoretic objectives of DisDiff cannot be naturally extended to a varying number of permutable objects, DisDiff is trained suboptimally and fails to disentangle both style and objects. In object disentanglement, object-centric approaches (LSD, L2C) show reasonable performance thanks to the slot-attention module. However, they completely fail in disentangling style information (∼10% accuracy.) This is because slot-attention lacks motivation for the disentanglement of spatially non-exclusive factors. This clear contrast highlights the strength of our unified approach in jointly disentangling multiple factors simultaneously without altering objectives or model architectures.

| | | | Shape3D | | | CLEVR | |
|---|---|---|---|---|---|---|---|
| | | | FactorVAE | DCI | Shape↑ | Color↑ | Position↓ |
| $\mathcal{L}_{\text{Diff}}$ | $\mathcal{L}_{\text{Prior}}$ | $\mathcal{L}_{\text{Con}}$ | | | *Impact of Losses* | | |
| ✓ | ✗ | ✗ | 0.492 | 0.175 | 62.27 | 88.58 | 0.111 |
| ✓ | ✓ | ✗ | 0.597 | 0.224 | 63.39 | 86.94 | 0.126 |
| ✓ | ✗ | ✓ | 0.769 | 0.597 | 64.21 | 80.28 | 0.116 |
| ✓ | ✓ | ✓ | **1.000** | **0.887** | **87.04** | **93.93** | **0.032** |
| Mixing strategy | | | | | *Impact of Mixing Strategy* | | |
| Attribute | | | **1.000** | **0.887** | 65.24 | 80.52 | 0.119 |
| Object | | | 0.634 | 0.127 | **87.04** | **93.93** | **0.033** |

Table 5: Ablation study on our method. It confirms that ours works best with all objectives and the proper mixing strategy.

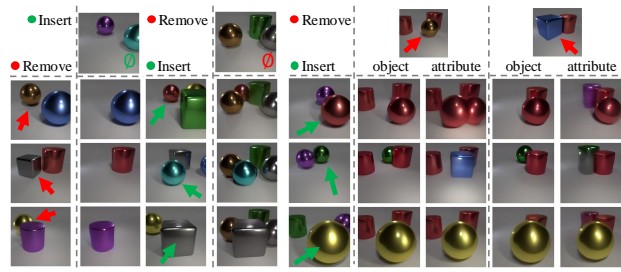

(a) OOD example 1    (b) OOD example 2    (c) Decoded images from different mixing strategy

Figure 7: Qualitative analysis on our method. It verifies OOD generalization (a),(b) and importance of mixing-strategy (c).

**Qualitative Results** In Fig. 5, we examined style transfer by replacing the latent representation encoding the style of target images with that of source images. While our method successfully transfers the source images' style into the target images, none of the baselines correctly modifies the overall style of the images. Although DisDiff often alters the style (first and third rows), it does not correctly reflect the original style and often changes the objects, indicating that the style is entangled with objects. We also present object-wise manipulation of our method in Fig. 6 on the right. Successful insertion and removal of the objects verify that our approach encodes individual objects in a disentangled way. It is worth noting that ours is the only method that disentangles both factors at the same time, and it is done by simply adjusting mixing strategies without altering objectives or model architectures. Additional results and analysis are provided in Appendix A.12.

## 4.4 Ablation Study

**Impact of Losses** In Tab. 5, we conduct an ablation study on each term of our objectives. The result show that all three losses ($\mathcal{L}_{\text{Diff}}$, $\mathcal{L}_{\text{Prior}}$, $\mathcal{L}_{\text{Con}}$) are essential. In attribute disentanglement, adding each loss term sequentially improves performance, with the best results when all losses are combined. For object disentanglement, clear improvements occur only when all loss terms are used together.

**Impact of Mixing Strategy** In the bottom three rows of Tab. 5, we investigate the importance of a proper mixing strategy for attribute and object disentanglement. We apply object mixing to attribute disentanglement and attribute mixing to object disentanglement, respectively. The results show that the interchanged mixing strategy significantly degrades performance in both attribute and object disentanglement, highlighting the importance of using the correct mixing strategy in our method.

**More qualitative analysis** In Fig 7-(a, b), we observe that our method is capable of generating out-of-distribution (OOD) examples that do not exist in the dataset, but can be created through composition. Notably, in the CLEVR-Easy dataset, which comprises images with 2-3 objects, our method can generate OOD images containing either a single object or 4 objects through composition, by inserting or removing the representation that does not encode the object. In Fig. 7-(c), we compare composite images from models trained with object mixing versus attribute mixing. Only the model trained with object mixing achieves object-wise manipulation, whereas the model trained with attribute mixing changes multiple objects simultaneously.

## 5 Conclusion

We introduced a modular inductive bias for DRL that is decoupled from both learning objectives and model architectures. We formulated DRL as maximizing data likelihood together with the compositional consistency of composite images produced by latent mixing, and embed factor-specific biases via a modular mixing strategy. This design enables attribute, object, and joint disentanglement within a single framework by adjusting only mixing strategy, without requiring changes in architectures or objective functions. Extensive evaluations demonstrate that our method surpasses state-of-the-art baselines in attribute disentanglement, while maintaining competitive performance in object disentanglement. More importantly, our method successfully disentangle both objects and global style at the same time in newly-introduced CLEVR-Style dataset, whereas current state-of-the-art DRL approaches fails to disentangle both factors.

**Acknowledgment** This work was in part supported by the National Research Foundation of Korea (RS-2024-00351212 and RS-2024-00436165) and the Institute of Information & communications Technology Planning & Evaluation (IITP) (RS-2022-II220926, RS-2022-II220959, RS-2024-00509279, and RS-2019-II190075) funded by the Korea government (MSIT).

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

# A   Appendix

## A.1   Limitations and Future Work

In our work, as in most previous approaches, we assume the target factors of variation are known in advance, and our goal is to learn how to disentangle them. However, in real-world scenarios, the factors of variation within a dataset may be unknown or considerably more complex than the idealized cases of attribute or object disentanglement. Consequently, an important future direction is to automatically identify underlying factors of variation and determine the appropriate mixing strategy (potentially through learning) without relying on prior knowledge beyond the dataset itself. Meanwhile, although our method shows strong performance in both attribute and object disentanglement, it does not provide theoretically grounded guarantees. Bridging the gap between methods that provide theoretical guarantees but only work on simple datasets, and methods like ours that demonstrate strong performance but lack such guarantees, is another important direction for future research. Moreover, although we focus on learning a disentangled representation in this work, exploring our framework on downstream tasks such as controllable image manipulation, *e.g.*, attribute- or object-level edits, and object-centric world-model training that leverages our object-disentangled representations, as exemplified by Jeong et al. [18].

## A.2   Broader Impact

Our method can extract attribute or object components from existing images and use this extracted information to generate new images. This capability may raise privacy issues if applied to deepfake generation or unauthorized copying of digital content.

## A.3   More Related Work

In this section, we discuss additional related work relevant to our method.

**Identifiable DRL**   In object-level disentangled representation learning (or object-centric learning), a line of work [2, 25, 50] leverages identifiability theory to derive conditions that provide theoretical guarantees for disentangled representations aligned with underlying factors of variation. Specifically, Brady et al. [2] shows that, under certain assumptions in object-centric scenes, an invertible *compositional decoder* can recover the ground-truth object latents (up to permutation). Lachapelle et al. [25] provides theoretical conditions for identifiability (up to permutation and blockwise invertible transformations) when ground-truth latent variables are organized into specific blocks and an *additive decoder* is used. While these work can provide identifiability guarantees for object representations, it may not generally apply to factors of variation such as attributes, which globally affect the image and do not necessarily satisfy the assumptions of additive decoders and compositionality definitions. Moreover, these approaches often impose strong restrictions on model design and expressiveness [41], making them applicable only to relatively simple datasets. Our method aims in a different direction from these works, seeking a modular inductive bias that is decoupled from both learning objectives and architectural constraints, such as an additive decoder, thereby enabling disentangled representation learning for various factors under a single objective and architecture.

**Wiedemer et al. [50]**   Wiedemer et al. [50] demonstrates that autoencoders satisfying encoder-decoder consistency, in combination with an additive decoder, can yield object-centric representations that provably generalize compositionally. While this approach shares the similar conceptual goal of enabling generalization to novel compositions of factors in disentangled representation learning as ours, there are fundamental differences in how we achieve valid generalization. As the common part, both methods recognize that valid generalization requires two key components: first, compositions of disentangled representations must yield valid data, *e.g.*, realistic images or representation, and second, the composite representation $\mathbf{z}^c$ must properly encode the information from its corresponding composite image $\mathbf{x}^c$ to satisfy the representation learning objective. To address the second requirement, both our method and Wiedemer et al. [50] employ a compositional consistency loss with minor differences. However, the approaches diverge significantly for the first component–how to ensure the validity of composite representations.

Wiedemer et al. [50] employs an additive decoder to ensure valid compositions of disentangled representations (Sec. 3.1 in [50]). However, additive decoders are known to be unscalable for complex scenes, as their local decoding mechanism cannot capture complex interactions between

Table 6: Comparison to Wiedemer et al. [50]. The additive decoder alone fails to achieve attribute, object, and joint disentanglement. While employing a slot attention module in the encoder leads to reasonable performance on object disentanglement, it still fails in joint disentanglement.

| | Shapes3D | | CLEVR | | | | CLEVR-Style | | | | |
|---|---|---|---|---|---|---|---|---|---|---|---|---|
| | FactorVAE | DCI | Shape | Color | Material | Position ($\downarrow$) | Acc | GRAM ($\downarrow$) | Shape | Color | Material | Position ($\downarrow$) |
| Ours | **0.975** | **0.837** | **87.04** | **93.93** | **94.81** | **0.032** | **96.50** | **5.05** | **83.56** | **90.48** | **93.74** | **0.053** |
| Additive Decoder w/o SA | 0.000 | 0.031 | 33.87 | 15.24 | 51.87 | 0.765 | 23.30 | 18.59 | 34.20 | 13.54 | 50.90 | 0.517 |
| Additive Decoder w/ SA | - | - | 82.91 | 93.14 | 91.78 | 0.110 | 23.30 | 15.84 | 35.97 | 20.99 | 54.10 | 0.520 |

objects due to limited expressive power. More critically, additive decoders are designed based on the spatial exclusiveness bias (Def. 4 in [50]), which assumes that each pixel should be affected by only a single latent variable. This limits their applicability to object-centric learning scenarios, preventing generalization to other disentanglement tasks.

In contrast, our method ensures validity through a prior loss (Eq. 4) that leverages SDS loss [34] to encourage composite image $\mathbf{x}^c$ to be realistic. As we do not require an additive decoder anymore, our approach allows us to utilize an expressive diffusion decoder for modeling complex scenes. Crucially, since our method does not embed factor-specific biases like spatial exclusiveness into the architecture, it can be applied beyond object-centric learning to attribute disentanglement and joint disentanglement scenarios. More importantly, to the best of our knowledge, we are the first to demonstrate that different underlying factors of variation can be disentangled simply by adjusting the factor-specific mixing strategy, distinguishing our approach from previous methods that introduce factor-specific biases through architectures or objective functions.

Finally, we provide a quantitative comparison of Wiedemer et al. [50] with our method in attribute-, object-, and joint disentanglement in Tab. 6. For a fair comparison, we used the same encoder for [50] as ours and only changed the decoder to an additive decoder. As expected, [50] cannot disentangle attribute factors at all (Shapes3D, CLEVR-Style), since they violate the spatial-exclusiveness assumptions. Moreover, in object disentanglement, we found that an additive decoder alone cannot disentangle objects (in fact, Wiedemer et al. [50] validated their method only on very simple 2D synthetic datasets). When we additionally use the slot-attention module in [50], it reasonably disentangles objects in the CLEVR dataset but is still significantly inferior to our method in CLEVR-Style, possibly due to the limited expressive power of the additive decoder. Additionally, we observed that object-wise manipulation with [50] always leads to unrealistic images with transparently overlapping objects due to the lack of interactions between latents inside the additive decoder.

**Group theory-based DRL**    Disentangled representation learning using Group theory is an actively researched area and is related to our work. Group theory-based DRL typically define disentangled representation $Z$ as follows: Given ground truth factors of variation $W$ and decomposable group $G$ (i.e., $G = G_1 \times G_2 \times \ldots \times G_n$), the representation $Z$ is disentangled w.r.t. $G$ if (1) there exists a mapping $f$ from $W$ to $Z$ such that $f(g \cdot W) = g \cdot f(W)$ for all $g \in G$ and $w \in W$, and (2) there is a decomposition $Z = Z_1 \times \ldots \times Z_n$ such that each $Z_i$ is affected only by $G_i$. Despite this convincing principled definition, since the GT factors of variation $W$ are infeasible to obtain in unsupervised DRL, existing unsupervised methods often utilize necessary conditions for group actions of $G$ applied to $Z$ and disentangled representation $Z$ [46, 53]. For instance, Tao et al. [46] defines permutation group actions of element-wise addition on $Z$ and introduces losses to enforce commutativity and cyclicity of group actions.

Our method takes a similar approach, but with important distinctions. From a Group theory perspective, unlike existing work, the group action in our method is defined on disentangled representation pairs $(z^1, z^2)$ rather than a single latent $z_i$. By defining the group action on a pair $(z^1, z^2)$, we can impose additional necessary conditions for how each underlying factor combines to generate observations, which cannot be induced by the group action defined on a single latent $z_i$. For example, commutativity and cyclicity of group action are necessary conditions for both attributes and objects, but do not impose attribute or object-specific properties. Our main contribution here is that we define group action as factor-specific mixing, *i.e.*, permutations, between two latents and demonstrate that this additional necessary condition imposes effective factor-specific inductive bias for attribute and object disentanglement without changing the overall learning objectives or model architectures. To the best of our knowledge, we are the first to study differently learned disentangled representations of

different factors of variation through a mixing strategy (or a form of group action) without employing factor-specific architectures or learning objectives.

**DRL with Factorized support**   Roth et al. [39] leverages a similar idea of combining latents to promote factorized support. While ours and prior work both leverage a factorized support assumption, our main contributions are fundamentally different. First of all, we demonstrate that additional assumptions about the factorization structure of support, *e.g.*, product of $K$ distinct supports in attributes or repetition of a shared support in objects, lead to disentanglement of different factors of variation. By combining these two assumptions, we can even achieve joint disentanglement of attributes and objects, which the factorized support assumption alone cannot handle. Secondly, we show that these factorization structures can be encoded via simple mixing strategies (Eqs. 1, 2, 3) replacing existing factor-specific biases. Those mixing strategies are decoupled from the architecture and objectives, so simply switching the mixing strategy enables us to disentangle attributes and objects within the single framework. Note that the prior method simply mixes the latent dimension-wise. Our ablations (Tab. 5) empirically show that such a strategy is insufficient for disentangling objects.

**Joint disentanglement of attribute and object**   Recently, SysBinder [43] introduced the Block-Slot representation, which models each object as a slot formed by concatenating multiple multi-dimensional attribute (factor) representations called blocks, thereby enabling disentangled representations of both objects and their attributes. This approach differs from ours by fully redesigning the model architecture to handle object-centric scenes, making it inapplicable to broader factors. In contrast, our method aims to support disentangled representation learning for various factors of variation within a single framework with a modular inductive bias. Nonetheless, extending our approach to disentangle multiple factors of variation using a modular inductive bias remains an important direction for future work.

### A.4   Equivalence between mixing two and multiple images

**Proof of equivalence**   In this section, we explain why the random mixing between two images (*i.e.*, $\mathbf{z}^c = \pi(\mathbf{z}^1, \mathbf{z}^2)$) can replace the random composition of $\mathbf{z}_i$ from $K$ images. Formally, we will show that:

$$\text{If} \quad \mathcal{S}(p(\mathbf{z})) = \mathcal{S}(p(\mathbf{z}^c)) \quad \text{then} \quad \mathcal{S}(p(\mathbf{z})) = \mathcal{S}^{\times}(p(\mathbf{z})), \tag{7}$$

where the factorized support $\mathcal{S}^{\times}(p(\mathbf{z})) = \mathcal{S}(p(\mathbf{z}_1)) \times \mathcal{S}(p(\mathbf{z}_2)) \times \cdots \times \mathcal{S}(p(\mathbf{z}_K))$ represents the random composition of each latent variable $\mathbf{z}_i$ from $K$ images.

*Proof.* Given $\mathcal{S}(p(\mathbf{z})) = \mathcal{S}(p(\mathbf{z}^c))$, we can prove the followings:

1. If $p(\mathbf{z}_1)p(\mathbf{z}_2) > 0$ then $p(\mathbf{z}_1, \mathbf{z}_2) > 0$.
   Note that $p(\mathbf{z}_1) > 0$ and $p(\mathbf{z}_2) > 0$ ($\Leftrightarrow p(\mathbf{z}_1)p(\mathbf{z}_2) > 0$) indicates the existence of $\mathbf{z}^1, \mathbf{z}^2$ with $\mathbf{z}_1^1 = \mathbf{z}_1, \mathbf{z}_2^2 = \mathbf{z}_2$. By mixing $\mathbf{z}^1$ and $\mathbf{z}^2$, we can compose $\mathbf{z}^*$ where $\mathbf{z}_1^* = \mathbf{z}_1, \mathbf{z}_2^* = \mathbf{z}_2$. Then, by the definition of the support that $\mathcal{S}(p(\mathbf{z})) = \{\mathbf{z}|p(\mathbf{z}) > 0\}$ and the given condition $\mathbf{z}^* \in \mathcal{S}(p(\mathbf{z}^c)) = \mathcal{S}(p(\mathbf{z})), p(\mathbf{z}_1, \mathbf{z}_2) \geq p(\mathbf{z}^*) > 0$.

2. Assume that for some $k \geq 2$, if $\prod_{i=1}^{k} p(\mathbf{z}_i) > 0 \rightarrow p(\mathbf{z}_1, \mathbf{z}_2, \ldots, \mathbf{z}_k) > 0$ then $\prod_{i=1}^{k+1} p(\mathbf{z}_i) > 0 \rightarrow p(\mathbf{z}_1, \mathbf{z}_2, \ldots, \mathbf{z}_k, \mathbf{z}_{k+1}) > 0$.
   Note that $\prod_{i=1}^{k+1} p(\mathbf{z}_i) > 0$ implies $p(\mathbf{z}_{k+1}) > 0$ and $\prod_{i=1}^{k} p(\mathbf{z}_i) > 0$. By the given assumption, $p(\mathbf{z}_1, \mathbf{z}_2, \ldots, \mathbf{z}_k) > 0$ and there exists $\mathbf{z}^1, \mathbf{z}^2$ where $\mathbf{z}_i^1 = \mathbf{z}_i$ for $i \in \{1, \ldots, k\}$ and $\mathbf{z}_{k+1}^2 = \mathbf{z}_{k+1}$. By mixing $\mathbf{z}^1$ and $\mathbf{z}^2$, we can compose $\mathbf{z}^*$ where $\mathbf{z}_i^* = \mathbf{z}_i$ for $i \in \{1, \ldots, k+1\}$. As a result, by the given condition $\mathbf{z}^* \in \mathcal{S}(p(\mathbf{z}^c)) = \mathcal{S}(p(\mathbf{z}))$, $p(\mathbf{z}_1, \mathbf{z}_2, \ldots, \mathbf{z}_k, \mathbf{z}_{k+1}) \geq p(\mathbf{z}^*) > 0$.

3. By mathematical induction, we conclude that if $\prod_{i=1}^{K} p(\mathbf{z}_i) > 0$ then $p(\mathbf{z}) > 0$.

Note that (3) implies $\mathcal{S}(p(\mathbf{z})) = \mathcal{S}^{\times}(p(\mathbf{z}))$, since $\mathcal{S}^{\times}(p(\mathbf{z}))$ can be expressed as $\{\mathbf{z}|p(\mathbf{z}_i) > 0\}$. By using mathematical induction, we have proved that random mixing between two images can replace the random composition of multiple images to achieve disentanglement.

**Empirical results**   In addition to proving equivalence, we compare our two image mixing strategies against mixing across multiple images (we use 64 here). We evaluate attribute disentanglement using

three random seeds and report the FactorVAE and DCI scores in Tab. 7. We observe no meaningful difference between mixing two images or 64 images, supporting our theoretical result.

Table 7: Effects of the number of samples used in mixing strategy.

| # of samples for mixing | FactorVAE | DCI |
|---|---|---|
| 2 | 0.975±0.040 | 0.837±0.105 |
| 64 | 0.966±0.032 | 0.802±0.088 |

## A.5 Evaluation metrics

In this section, we provide brief descriptions and definitions of the metrics we used in our experiments.

**FG-ARI** *Foreground Adjusted Rand Index* measures the agreement between predicted and ground-truth *instance* partitions, restricted to foreground pixels. Let $\mathcal{I}$ be the set of foreground pixels in ground-truth labels, and let $y \in \{1, \ldots, K\}^{|\mathcal{I}|}$ and $\hat{y} \in \{1, \ldots, \hat{K}\}^{|\mathcal{I}|}$ denote ground-truth and predicted instance labels on $\mathcal{I}$, respectively. Then, we define the contingency table $n_{ij} = |\{p \in \mathcal{I} : y_p = i, \hat{y}_p = j\}|$, row sums $a_i = \sum_j n_{ij}$, and column sums $b_j = \sum_i n_{ij}$, with $n = \sum_{ij} n_{ij} = |\mathcal{I}|$. The FG-ARI is defined as

$$\text{FG-ARI} = \frac{\sum_{i,j} \binom{n_{ij}}{2} - \frac{\sum_i \binom{a_i}{2} \sum_j \binom{b_j}{2}}{\binom{n}{2}}}{\frac{1}{2}\left(\sum_i \binom{a_i}{2} + \sum_j \binom{b_j}{2}\right) - \frac{\sum_i \binom{a_i}{2} \sum_j \binom{b_j}{2}}{\binom{n}{2}}},$$

which lies in $[-1, 1]$ and is higher when partitions agree better.

**mIoU** *Mean Intersection-over-Union* averages class-wise IoU. For class $c$, let $P_c$ and $G_c$ be predicted and ground-truth masks. Then mIoU is defined as

$$\text{IoU}_c = \frac{|P_c \cap G_c|}{|P_c \cup G_c|}, \qquad \text{mIoU} = \frac{1}{|\mathcal{C}|} \sum_{c \in \mathcal{C}} \text{IoU}_c,$$

where $\mathcal{C}$ is the set of evaluated classes.

**mBO** *Mean Best Overlap* measures, for each ground-truth object, how well it is covered by its single most overlapping prediction, and then averages these values. Let $\mathcal{O}$ be the set of ground-truth instances with masks $\{G_o\}_{o \in \mathcal{O}}$, and let $\{M_k\}_{k=1}^K$ be the set of predicted instance masks. For each $o$,

$$\text{BO}(o) = \max_{1 \le k \le K} \text{IoU}(G_o, M_k), \qquad \text{mBO} = \frac{1}{|\mathcal{O}|} \sum_{o \in \mathcal{O}} \text{BO}(o).$$

**FactorVAE Score [22]** The *FactorVAE Score* quantifies disentanglement of a representation with respect to generative factors. We repeatedly sample mini-batches in which exactly one factor is fixed and all others vary. For each batch, we compute the empirical variance of each latent coordinate, normalize coordinate-wise, *e.g.*, by the mean variance across coordinates, and select the index of the lowest-variance coordinate as a feature. A simple classifier, such as a majority vote, is used to predict the fixed factor index from that feature. The classification accuracy on held-out batches is reported as the FactorVAE Score (higher is better).

**Disentanglement Score in DCI [10]** Let $R \in \mathbb{R}^{d \times M}$ be an importance matrix obtained by training a predictive model from latent coordinates $z \in \mathbb{R}^d$ to factors $\{v_m\}$, where $R_{jm} \ge 0$ measures the contribution of latent dimension $j$ to predicting factor $m$. Define normalized importances $\rho_{jm} = R_{jm}/\sum_{m'} R_{jm'}$. The per-dimension disentanglement is

$$D_j = 1 - H(\rho_{j:})/\log M,$$

where $H$ is the entropy. Weighting by total importance $w_j = \sum_m R_{jm}$ and normalizing, the overall DCI Disentanglement is

$$\text{DCI-D} = \sum_{j=1}^d \tilde{w}_j D_j, \quad \tilde{w}_j = \frac{w_j}{\sum_{j'} w_{j'}}.$$

**GRAM Loss** A *Gram loss* measures feature correlations of a neural network $\phi$ at selected layers $\mathcal{L}$. It measures second-order feature statistics (style) between two images $x$ and $y$. For two images $x$ and $y$, let $F_\ell(x) \in \mathbb{R}^{C_\ell \times H_\ell W_\ell}$ be the feature at layer $\ell$, obtained by reshaping $\phi_\ell(x)$. The Gram matrix is $G_\ell(x) = \frac{1}{C_\ell H_\ell W_\ell} F_\ell(x) F_\ell(x)^\top$. The GRAM loss is measured as

$$\mathcal{L}_{\mathrm{GRAM}}(x,y) = \sum_{\ell \in \mathcal{L}} \|G_\ell(x) - G_\ell(y)\|_F^2 \,.$$

## A.6 Additional Implementation Details

When training our model, we use a fixed batch size of 64 and a learning rate of 0.0001 across all of the experiments. We use $\lambda_{\mathrm{Prior}} = 1$ and $\lambda_{\mathrm{Con}} = 0.01$ for all experiments. We set the number of latents to $k = 10$ in attribute disentanglement and use $K = 4, 11, 11, 12, 6$ for CLEVREasy, CLEVR, CLEVRTex, ClevrTex-Style, MSN-Style datasets in object disentanglement, respectively. When training the diffusion model, we use a v-prediction [40] loss to ensure reliable few-step generation.

Tab. 8– 17 summarizes the hyper-parameters of our encoder and decoder architectures used in the experiments. Following the diffusion-based baselines: DisDiff [52] and LSD [19], we employ pre-trained VQ-VAE [3] and KL-regularized auto-encoder model [4] in attribute and object disentanglement, respectively. In attribute disentanglement, the encoder first maps the input $\mathbf{x}$ into a $KD$-dimensional vector $\mathbf{z} \in \mathbb{R}^{KD}$ and then uniformly divides it into $K$ latents. In object disentanglement, we adopt the Q-former [27] of 4 transformer blocks with 8 attention heads and a hidden dimension of 256. Specifically, we have $K$ learnable queries $\{\mathbf{q}\}^K \in \mathbb{R}^{K \times D}$ and those queries are updated via multiple self attention layers and cross attention layers, where the keys and values are linearly projected from the U-net encoder feature of $\mathbf{x}$.

---

[3]     https://huggingface.co/stabilityai/sd-vae-ft-ema-original
[4]     https://ommer-lab.com/files/latent-diffusion/celeba.zip

```
Conv 3 × 3 × 3 × 128, stride=1
BatchNorm2d
ReLU
Conv 3 × 3 × 128 × 128, stride=1
BatchNorm2d
ReLU
Conv 3 × 3 × 128 × 128, stride=1
BatchNorm2d
ReLU
Conv 3 × 3 × 128 × 128, stride=1
BatchNorm2d
ReLU
ResBlock 3 × 3 × 128 × 128, stride=1
BatchNorm2d
ReLU
ResBlock 3 × 3 × 128 × 128, stride=1
BatchNorm2d
ReLU
FC 4096 × 10
```

Table 8: Encoder Architecture used in attribute disentanglement.

```
ReLU
Conv 3 × 3 × 128 × 128, stride=1
BatchNorm2d
ReLU
Conv 3 × 3 × 128 × 128, stride=1
```

Table 9: ResBlock in the Encoder

| | |
|---|---|
| Input Resolution | 16 |
| Input Channels | 3 |
| Input Channels | 4 |
| $\beta$ scheduler | Linear |
| Mid Layer Attention | Yes |
| # Res Blocks / Layer | 2 |
| # Heads | 8 |
| Base Channels | 64 |
| Attention Resolution | [1,2,4,4] |
| Channel Multipliers | [1,2,4,4] |

Table 10: Decoder Architecture used in attribute disentanglement

| | |
|---|---|
| Input Resolution | 16 |
| Input Channels | 3 |
| Output Resolution | 16 |
| Self Attention | Middle Layer |
| Base Channels | 128 |
| Channel Multipliers | [1,1,2,4] |
| # Heads | 8 |
| # Res Blocks / Layer | 1 |

Table 11: Unet Encoder Architecture used in object disentanglement.

| | |
|---|---|
| Input Resolution | 16 |
| Input Channels | 4 |
| $\beta$ scheduler | Linear |
| Mid Layer Attention | Yes |
| # Res Blocks / Layer | 2 |
| # Heads | 8 |
| Base Channels | 192 |
| Attention Resolution | [1,2,4,4] |
| Channel Multipliers | [1,2,4,4] |

Table 12: Decoder Architecture used in object disentanglement.

| | |
|---|---|
| Input Resolution | 16 |
| Input Channels | 3 |
| $\beta$ scheduler | Linear |
| Mid Layer Attention | Yes |
| # Res Blocks / Layer | 2 |
| # Heads | 8 |
| Base Channels | 64 |
| Attention Resolution | [1,2,4,4] |
| Channel Multipliers | [1,2,4,4] |

Table 13: Generative Prior Architecture used in attribute disentanglement.

| | |
|---|---|
| Input Resolution | 16 |
| Input Channels | 4 |
| $\beta$ scheduler | Linear |
| Mid Layer Attention | Yes |
| # Res Blocks / Layer | 2 |
| # Heads | 8 |
| Base Channels | 192 |
| Attention Resolution | [1,2,4,4] |
| Channel Multipliers | [1,2,4,4] |

Table 14: Generative Prior Architecture used in object disentanglement.

```
Conv 3 × 3 × 3 × 128, stride=1
BatchNorm2d
ReLU
Conv 3 × 3 × 128 × 128, stride=1
BatchNorm2d
ReLU
Conv 3 × 3 × 128 × 128, stride=1
BatchNorm2d
ReLU
Conv 3 × 3 × 128 × 128, stride=1
BatchNorm2d
ReLU
ResBlock 3 × 3 × 128 × 128, stride=1
BatchNorm2d
ReLU
ResBlock 3 × 3 × 128 × 128, stride=1
BatchNorm2d
ReLU
FC 4096 × 10
```

Table 15: Encoder Architecture used in attribute disentanglement.

```
ReLU
Conv 3 × 3 × 128 × 128, stride=1
BatchNorm2d
ReLU
Conv 3 × 3 × 128 × 128, stride=1
```

Table 16: ResBlock in the Encoder

| | |
|---|---|
| Input Resolution | 16 |
| Input Channels | 4 |
| $\beta$ scheduler | Linear |
| Mid Layer Attention | Yes |
| # Res Blocks / Layer | 2 |
| # Heads | 8 |
| Base Channels | 192 |
| Attention Resolution | [1,2,4,4] |
| Channel Multipliers | [1,2,4,4] |

Table 17: Generative Prior Architecture used in object disentanglement.

### A.7 Details on Construction of CLEVR-Style and MSN-Style datasets

To construct the CLEVR-Style dataset, we first sample 25K images from the original CLEVR dataset and then augment each with three additional styles. Including the unmodified images, this produces a total of 80K/10K/10K images for the train/val/test splits, respectively. All augmentations combine simple color-space adjustments with diffusion-based translation. The first style (second column of Fig. 8) applies an HSV shift (hue=0, saturation=8, value=2) followed by image translation using Stable Diffusion [38] with the prompt "an oil painting." The second style (third column of Fig. 8) applies an HSV shift (hue=0, saturation=1.5, value=2.5), converts to LAB color space, and reduces contrast by a factor of 0.15. The third style (fourth column of Fig. 8) applies an HSV shift (hue=5, saturation=3, value=1), converts to LAB color space for CLAHE (Contrast Limited Adaptive Histogram Equalization), and then segments the image into 480 superpixels. These combined transforms produce 4 different visual styles while preserving the underlying scene composition.

Similarly, we construct the MSN-Style dataset by first sampling 15k images from the original MSN dataset [45] and augmenting with three identical styles used in CLEVR-Style. It produces a total of 40k/10k/10k images for the train/val/test splits, respectively. Fig. 9 presents the sample in MSN-Style with four different styles.

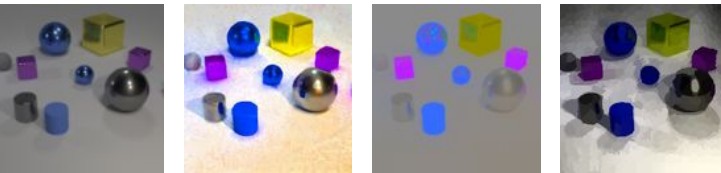

Figure 8: Example of CLEVR-Style dataset

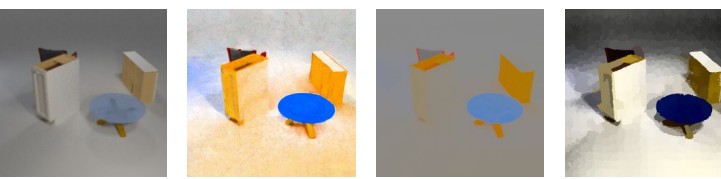

Figure 9: Example of MSN-Style dataset

### A.8 Experimental Details for Object Property Prediction

**Matching Technique**    We developed a technique to identify the specific region corresponding to an object's representation by analyzing images composed with that representation. For a given target object latent representation, we first randomly sample multiple images and encode each into an object representation. For each image, we then replace one object latent with the target latent, then decode the mixed representations. If the target object is properly encoded, it appears in the generated images. To determine the object region, we measure the RGB variance across these generated images and compare each generated image to the original one containing the target object representation. Finally, we combine these two metrics: variance and distance to the original image, to specify the object's region. We provide the pseudocode in Algorithm 1.

**Evaluation protocol for object property prediction**    For each property, we follow [21] to train a two-layer MLP classifier with hidden dimension 256 on the frozen object representations.

---

**Algorithm 1** Matching Technique

---

**Require:** $x$: an image; $z = \text{enc}(x)$: object representation of $x$; $n$: number of latent vectors; $x_{\text{ref}}$: randomly sampled reference $B$ images
1: **function** MATCHING($z, x, x_{\text{ref}}$)
2:  $\quad z_{\text{ref}} \leftarrow [\,\text{encode}(\_x) \text{ for } \_x \in x_{\text{ref}}\,]$
3:  $\quad z_{\text{mixed}} \leftarrow [\,\text{replace\_ith\_latent}(z_{\text{ref}}, z, i) \text{ for } i = 1 \dots n\,]$
4:  $\quad x_{\text{mixed}} \leftarrow [\,\text{decode}(\_z) \text{ for } \_z \in z_{\text{mixed}}\,]$
5:  $\quad x_{\text{ref\_cm}} \leftarrow \text{mean}(x_{\text{ref}}, \text{dim} = 0)$
6:  $\quad x_{\text{mixed\_cm}} \leftarrow \text{mean}(x_{\text{mixed}}, \text{dim} = 1)$
7:  $\quad s \leftarrow \text{softmax}(1 - \text{distance}(x, x_{\text{mixed\_cm}}).\text{mean}(-1))$
8:  $\quad d \leftarrow \text{softmax}(\text{distance}(x_{\text{mixed\_cm}}, x_{\text{ref\_cm}}).\text{mean}(-1))$
9:  $\quad$ **return** $(s + d).\text{argmax}(\text{dim} = 0)$     ▷ Mask indicating matched region
10: **end function**

---

### A.9 Comparison to Likelihood Maximization by Jung et al. [21]

We conducted experiments on the CLEVRTex dataset by replacing our prior loss with the likelihood maximization loss proposed in L2C. Tab. 18 shows the results, which clearly demonstrate the superior performance of our likelihood maximization term compared to L2C.

Table 18: Object property prediction results with L2C's likelihood maximization (prior) loss

|  | shape ($\uparrow$) | material ($\uparrow$) | position ($\downarrow$) |
| --- | --- | --- | --- |
| Ours | 70.90 | 52.20 | 0.133 |
| Ours + L2C prior loss | 54.58 | 27.18 | 0.165 |

### A.10 Additional Results on Attribute Disentanglement

As in the Shapes3D dataset, our method captures three independent factors, *i.e.*, direction, axis, and appearance, enabling controlled manipulation of each factor in the Cars3D dataset and presents the result in Fig. 10.

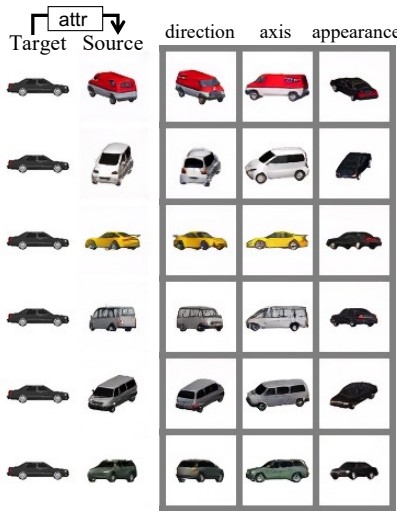

Figure 10: Qualitative results on attribute disentanglement in Cars3D dataset.

### A.11 Additional Details and Results on Unsupervised Object Segmentation

As described in the main paper, our method does not include a built-in mechanism (*e.g.*, slot-attention) to explicitly cluster pixels. To address this, we followed [31] to train the Spatial Broadcast Decoder (SBD)[49] on the frozen latent representations to predict object masks for each latent. Specifically, each frozen object representation is decoded individually by the SBD into an RGB image and an alpha mask. We then normalize the alpha masks across all object representations using a softmax and use them as mixture weights to combine the RGB outputs into a single image. We treat the normalized alpha masks as object-mask proxies for evaluation. The decoder is trained with a reconstruction loss to recover the original image for 30k iterations with a learning rate of 0.001. Because the encoder is frozen and the SBD is shallow (see Tab. 19), this training process is relatively inexpensive.

While conventional methods often extract object masks from the attention weights of slot-attentions, we evaluate the baselines using both these slot-attention masks and masks obtained by training an SBD on their frozen latent representations, for a fair comparison. The full results are reported in Tab. 20, Tab. 21. In the main paper, we present only the SBD-based results, since they outperform those derived from slot-attention masks. In Fig. 11, we also present object segmentation results on CLEVR and CLEVRTex datasets.

Table 19: Decoder Architecture used in object segmentation

| |
|---|
| Deconv $5 \times 5 \times 64 \times 64$, stride=2, padding=2, output_padding=1 |
| ReLU |
| Deconv $5 \times 5 \times 64 \times 64$, stride=2, padding=2, output_padding=1 |
| ReLU |
| Deconv $5 \times 5 \times 64 \times 64$, stride=2, padding=2, output_padding=1 |
| ReLU |
| Deconv $5 \times 5 \times 64 \times 64$, stride=2, padding=2, output_padding=1 |
| ReLU |
| Deconv $5 \times 5 \times 64 \times 64$, stride=1, padding=1, output_padding=1 |
| ReLU |
| Deconv $5 \times 5 \times 64 \times 4$, stride=1, padding=1, output_padding=1 |
| ReLU |

Table 20: Quantitative Results of unsupervised segmentation in the CLEVR dataset.

| Method | FG-ARI | | mIoU | | mBO | |
|---|---|---|---|---|---|---|
| | Slot-Attention | SBD Mask | Slot-Attention | SBD Mask | Slot-Attention | SBD Mask |
| LSD | 82.00 | **91.74** | 22.69 | 25.59 | 22.98 | 25.84 |
| L2C | 54.01 | 80.05 | 19.30 | 25.61 | 20.36 | 26.33 |
| Ours | - | 91.20 | - | **26.54** | - | **26.65** |

Table 21: Quantitative Results of unsupervised segmentation in CLEVRTex dataset.

| Method | FG-ARI | | mIoU | | mBO | |
|---|---|---|---|---|---|---|
| | Slot-Attention | SBD Mask | Slot-Attention | SBD Mask | Slot-Attention | SBD Mask |
| LSD | 46.54 | 71.64 | 45.87 | 56.26 | 46.93 | 56.75 |
| L2C | 77.07 | 82.55 | 56.59 | 58.33 | 53.25 | 58.68 |
| *Ours* | - | **87.68** | - | **58.88** | - | **59.12** |

Figure 11: Unsupervised object segmentation results in the CLEVR and CLEVRTex dataset. In CLEVRTex, our method consistently encodes complete objects into distinct latents, resulting in clean object segmentation results. On CLEVR, the constant background appears in the object regions of the segmentation results, leading to relatively lower mIoU and mBO (Tab. 20). However, this does not affect the underlying quality of object representations, as our method still consistently captures each entire object in its segmentation region.

## A.12 Additional Results on Joint Disentanglement of Attribute and Object

We present additional qualitative results on joint disentanglement of attribute (style) and object in Fig. 12– 15 and Fig. 16, respectively. While all the baselines struggle to disentangle the style information into a single latent representation, our method successfully disentangles the style and transfers it from source to target images. In addition to style disentanglement, our method also disentangles individual objects and enables object-wise manipulation as shown in Fig. 16.

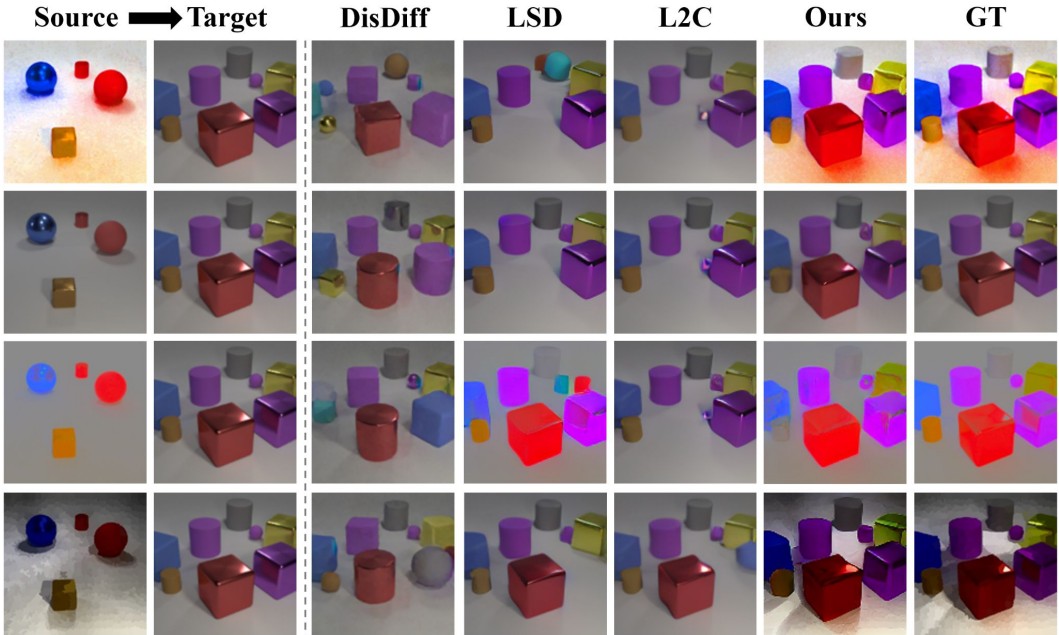

Figure 12: Style Transfer in CLEVR-Style dataset.

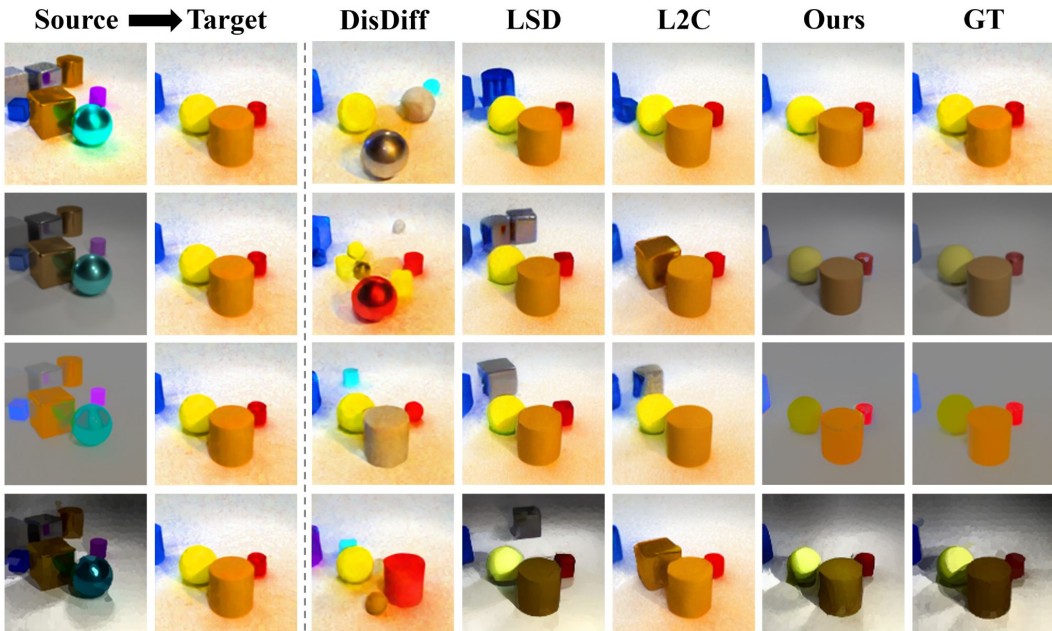

Figure 13: Style Transfer in CLEVR-Style dataset.

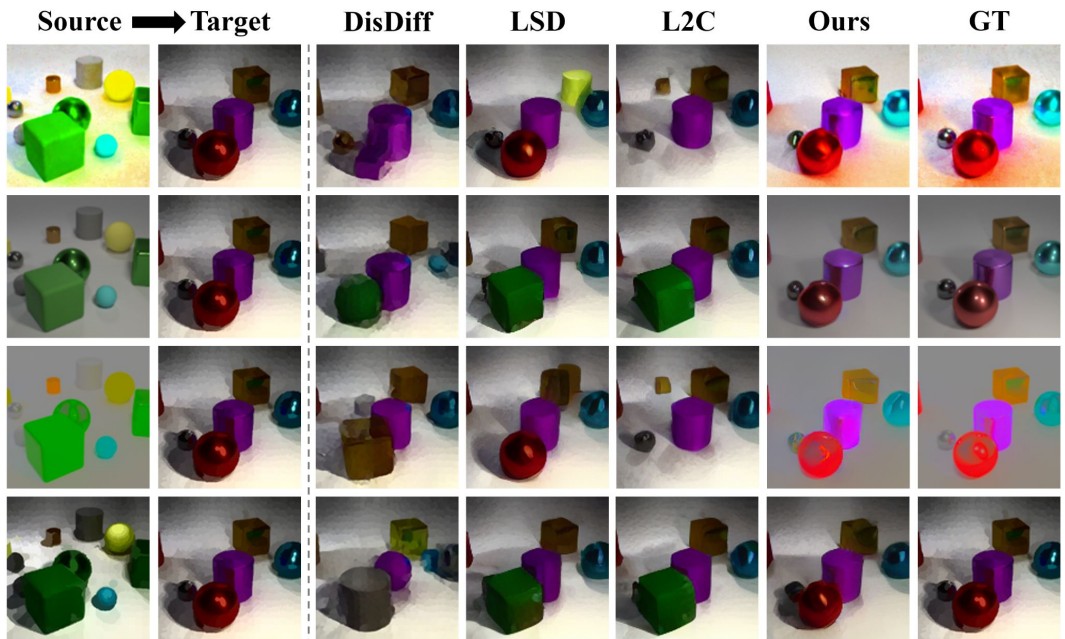

Figure 14: Style Transfer in CLEVR-Style dataset.

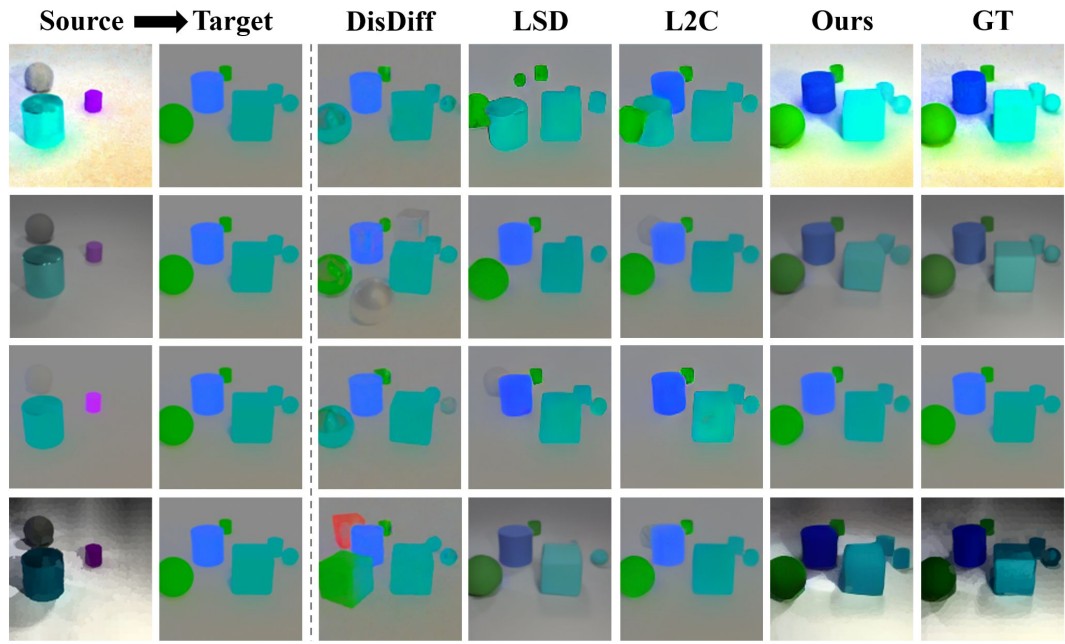

Figure 15: Style Transfer in CLEVR-Style dataset.

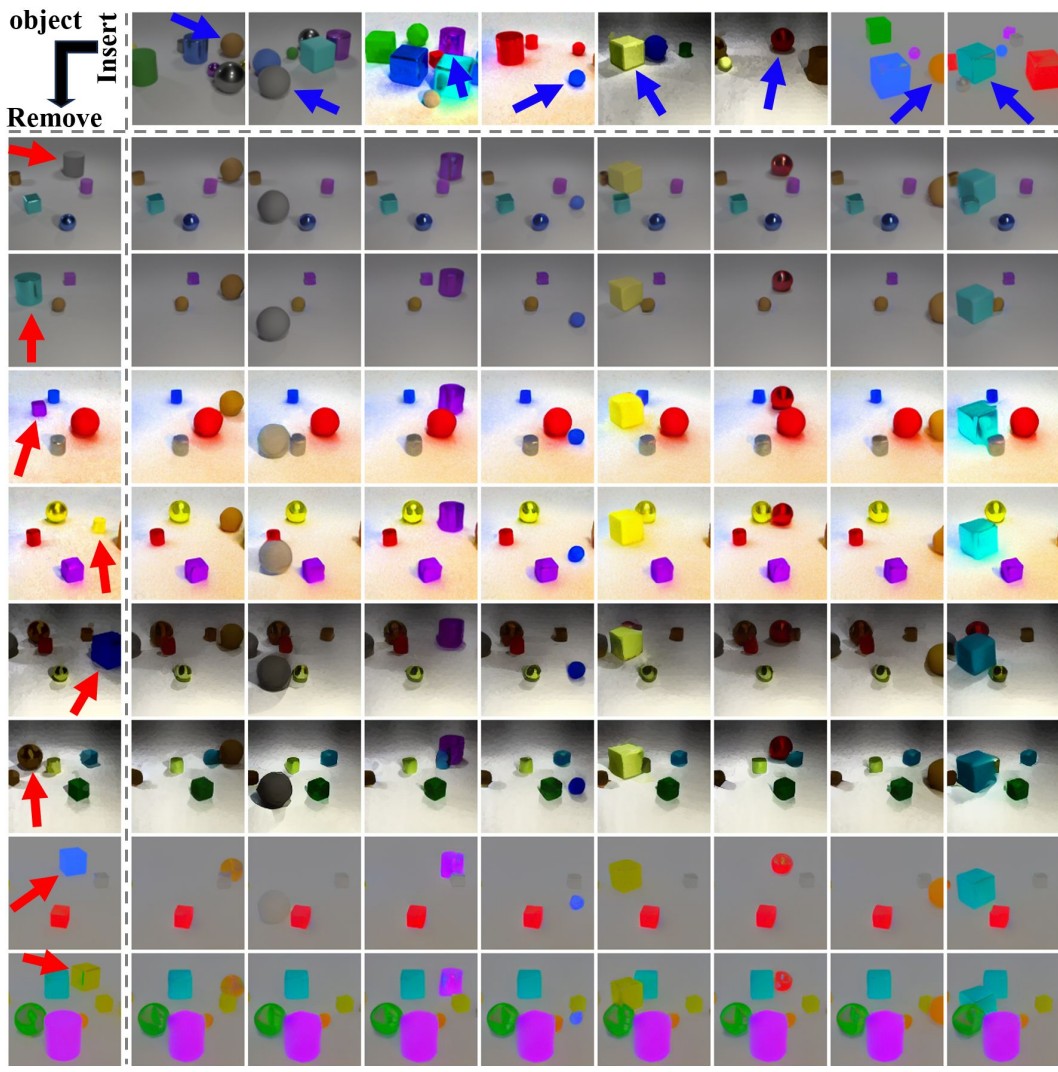

Figure 16: Additional qualitative results on Object Manipulation in CLEVR-Style dataset. Objects marked by red arrows are replaced with those marked by blue arrows. It demonstrates that our method effectively disentangles individual objects.

### A.13 Additional Results of Joint Disentanglement on Complex Dataset

To further validate our method on more complex and realistic datasets, we augment the MultiShapeNet (MSN) dataset [45] with four different global styles as in our previous CLEVR-Style experiments (See Appendix A.7 for details). MSN includes 11,733 unique furniture shapes with increased visual complexity compared to CLEVR. We compare our method with the strongest attribute-disentanglement baseline (DisDiff) and object-disentanglement baselines (LSD, L2C), reporting quantitative results in Tab. 22.

For style disentanglement, our method achieved the highest style prediction accuracy (Acc) and the lowest style loss (GRAM), demonstrating that it successfully isolates style information in a single latent and transfers it faithfully to the target image. Since baselines lack an internal mechanism to specify the latent representation for style information, we used a simple trick to identify the latent representation encoding style information. We decode all $K \times K$ possible latent exchanges between two sets of $K$ latents extracted from two images, and choose the pair yielding the lowest GRAM loss. Even with their best-case result, they fell short of our performance, confirming that they do not reliably capture a style factor. Although L2C achieves relatively high style prediction accuracy, style-swapping results in Fig. 17, 18 shows that swapping the single latent representation affects both the global style and objects, which indicates that L2C fails to isolate the global style but rather is entangled with object information. DisDiff also fails to produce reasonable compositions, and we conjecture that this is due to their disentanglement objective (variant/invariant loss), which is not well-suited for multi-object scenes and destabilizes the overall training.

For object disentanglement, Tab. 22 shows that our method produces the highest mIoU and mBO, indicating accurate localization and tight boundaries for each object mask, while FG-ARI was a bit lower than that of LSD and L2C. This trade-off arises because our broadcast decoder generated tight, object-internal masks. FG-ARI—which measures membership between two pixels within the same group—penalizes such masks when GT segmentations are looser, whereas mIoU and mBO reward the improved boundary precision. In contrast, slot attention modules in LSD and L2C often generate larger masks that bleed into the background or even include other objects, which leads to higher FG-ARI at the expense of mIoU and mBO. DisDiff, whose information-theoretic objective is not designed for a varying number of objects, fails to effectively separate either style or object factors.

Table 22: Quantitative results on Joint Disentanglement in MSN-Style

| Method | Style | | Object | | |
|--------|-------|-----|--------|-----|-----|
| | Acc(↑) | GRAM(↑) | FG-ARI(↑) | mIoU (↑) | mBO (↑) |
| DisDiff | 37.0 | 10.8 | 5.62 | 7.69 | 9.13 |
| LSD | 12.5 | 11.3 | 52.7 | 23.3 | 23.9 |
| L2C | 81.0 | 8.35 | **53.3** | 28.4 | 28.8 |
| Ours | **97.5** | **7.16** | 42.3 | **44.1** | **44.2** |

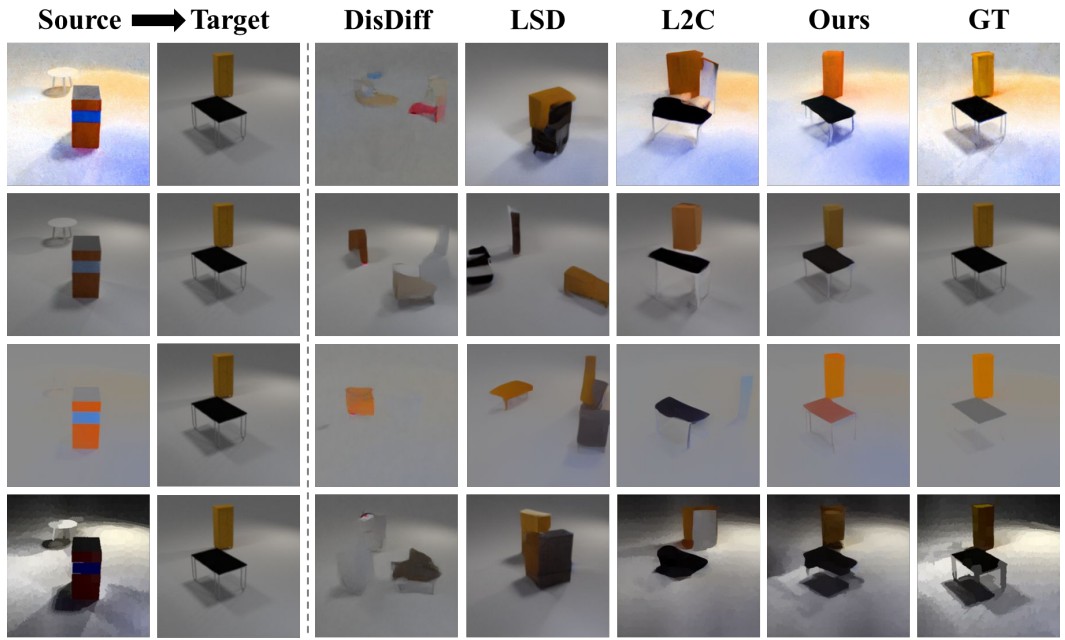

Figure 17: Style transfer results on MSN-Style dataset.

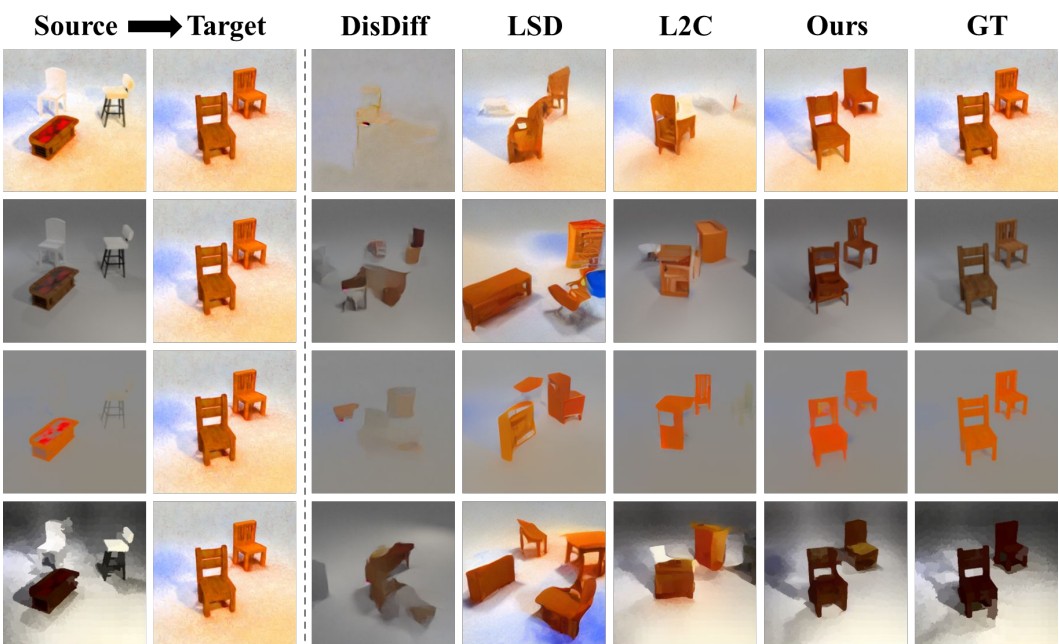

Figure 18: Style transfer results on MSN-Style dataset.

## A.14 Effect of random seeds on performance

We repeat our attribute/object disentanglement experiments with ten/three different random seeds and present the results in Tab. 23, Tab. 24, respectively, showing that our method achieves competitive performance in both tasks.

Table 23: Quantitative results on scene-level disentanglement. Our method achieves state-of-the-art performance in almost all of the datasets, except FactorVAE score in Cars3D.

| Method | Cars3D | | Shapes3D | | MPI3D | |
|---|---|---|---|---|---|---|
| | FactorVAE | DCI | FactorVAE | DCI | FactorVAE | DCI |
| FactorVAE [22] | 0.906±0.052 | 0.161±0.019 | 0.840±0.066 | 0.611±0.082 | 0.152±0.025 | 0.240±0.051 |
| β-TCVAE [5] | 0.855±0.082 | 0.140±0.019 | 0.873±0.074 | 0.613±0.114 | 0.179±0.017 | 0.237±0.056 |
| InfoGAN-CR [29] | 0.411±0.013 | 0.020±0.011 | 0.587±0.058 | 0.478±0.055 | 0.439±0.061 | 0.241±0.056 |
| LD [47] | 0.852±0.039 | 0.216±0.072 | 0.805±0.064 | 0.380±0.064 | 0.391±0.039 | 0.196±0.038 |
| GS [15] | 0.932±0.018 | 0.209±0.031 | 0.788±0.091 | 0.284±0.034 | 0.464±0.036 | 0.229±0.042 |
| DisCo [37] | 0.855±0.074 | 0.271±0.037 | 0.877±0.031 | 0.708±0.048 | 0.371±0.030 | 0.292±0.024 |
| DisDiff-VQ [52] | **0.976±0.018** | 0.232±0.019 | 0.902±0.043 | 0.723±0.013 | 0.617±0.070 | 0.337±0.057 |
| **Ours** | 0.877±0.089 | **0.365±0.073** | **0.975±0.059** | **0.837±0.105** | **0.708±0.060** | **0.458±0.052** |

Table 24: Object property prediction results with 3 different runs for our model.

| Method | CLEVREasy | | | CLEVR | | | | CLEVRTex | | |
|---|---|---|---|---|---|---|---|---|---|---|
| | Shape (↑) | Color (↑) | Position* (↑) | Shape (↑) | Color (↑) | Material (↑) | Position (↓) | Shape (↑) | Material (↑) | Position (↓) |
| SA | 72.25 | 72.33 | 44.08 | 79.4 | 91.30 | 93.18 | 0.064 | 30.44 | 7.890 | 0.482 |
| SLASH | 86.06 | 89.23 | 46.97 | 83.28 | 92.26 | 93.16 | 0.078 | 53.13 | 37.49 | 0.148 |
| LSD | **96.03** | **98.05** | 50.29 | **87.66** | 91.46 | **94.87** | 0.062 | 68.25 | 51.54 | 0.197 |
| L2C | 92.78 | 93.57 | 47.62 | 73.61 | 74.03 | 86.93 | 0.168 | **71.54** | 51.62 | **0.116** |
| **Ours** | 93.74±2.10 | 94.29±0.97 | 49.42±1.15 | 85.72±0.37 | **93.79±0.22** | 94.93±0.07 | **0.058±0.006** | 68.29±2.55 | 47.89±4.89 | 0.143±0.009 |

## A.15 Experiments on Correlated Factors

The datasets used in our main experiments handle only simple scenarios with statistically independent factors of variation (FoVs), unlike real-world cases. To verify our method on more challenging scenarios with correlated factors, we followed Roth et al. [39] to generate correlated benchmarks. We introduced 0.1 correlation between 1/2/3 pairs of GT factors in Shapes3D, as in [39], then trained and evaluated our models using FactorVAE and DCI metrics across 3 seeds. Tab. 25 reports the results.

Our method showed only slight metric drops despite correlations. Unlike prior work that relies on statistical independence between latent variables, *e.g.*, Total Correlation, our compositional objective only encourages discovering atomic factors whose compositions are valid under predefined mixing strategies. This allows successful disentanglement even with correlated factors, as the compositional requirement doesn't penalize correlations.

Table 25: Quantitative results on the dataset with correlated factors. Our method robustly disentangles the underlying ground-truth factors even with correlated factors.

| Amount of Correlation | Factor VAE | DCI |
|---|---|---|
| No Correlation | 0.975±0.059 | 0.837±0.105 |
| Pairs: 1, $\sigma = 0.1$ | 0.930±0.061 | 0.798±0.082 |
| Pairs: 2, $\sigma = 0.1$ | 0.997±0.002 | 0.806±0.060 |
| Pairs: 3, $\sigma = 0.1$ | 0.965±0.030 | 0.801±0.024 |

## A.16 Computing Resources

We conduct all our experiments on a GPU Server that consists of an Intel Xeon Gold 6230 CPU, 256GB RAM, and 8 NVIDIA RTX 3090 GPUs (with 24GB VRAM), or 8 NVIDIA RTX 6000 GPUs (with 48GB VRAM). It takes about 24 GPU hours and from 36 to 48 GPU hours for the attribute and object disentanglement experiment, respectively.

