# OpenReview forum: "Disentangled Representation Learning via Modular Compositional Bias"
_NeurIPS.cc/2025/Conference — NeurIPS 2025 poster_

### Official Review · Reviewer_XmPJ · 2025-06-23

**Clarity:** 3
**Significance:** 3
**Originality:** 3
**Rating:** 5
**Confidence:** 4

**Summary:**

This paper proposes a novel, general method for disentangled representation learning (DRL), with the lens of (attribute and object) compositionality. The proposed compositional bias does not rely on leveraging specific attribute properties (such as, e.g., the geometry of the factors) and, thus, is very flexible. This is demonstrated by extensive experiments.

**Questions:**

1. Do you have an intuition why your method works worse on Cars3D?
2. Could you please spell out with formulas/pseudocode your matching technique from A.7?
3. For your baseline models, how much hyperparameter tuning did you do?

**Ethical Concerns:**

["NO or VERY MINOR ethics concerns only"]

**Final Justification:**

Empirical and methodological contributions are useful and meaningful,  and the authors resolved my concerns regarding clarifying their contributions.

**Limitations:**

Yes

**Quality:**

3

**Strengths And Weaknesses:**

## Strengths
- The paper is well written, it's high-level message easy to follow (thanks to Fig. 1 and the contribution list)
- The experiments are extensive (and show superior performance or comparable to SOTA)
- The authors address their works limitations (no theoretical guarantees, no handling of unknown factors)

## Weaknesses

### Major points
**My main concern is the similarity (and even the same naming) to Wiedemer et al., 2024, which is not discussed in the paper (it is reference number 45, which is only briefly mentioned in the appendix, without a substantial discussion**. Especially that said paper has theoretical guarantees. I do notice that this paper has more extensive experimental evidence, and the formulation of the mixing strategy seems to be novel; however, that does not eliminate the need of the comparison to Wiedemer et al., 2024. I consider addressing this point as the necessary condition for the re-evaluation of my score.

- **Abstract:** even though I work on compositional generalization, I could not really figure out the contributions from the abstract (particularly the mixing strategies part). After reading the paper, it became clear. I'd suggest rephrasing it (I am happy to iterate over suggestions during the rebuttal)
- **Object and attribute latents:** please distinguish in the notation of the two types of latents better (I get that the super-/subscipts aim to indicate that). Particularly, I believe (2) might be incorrect: you cannot pick any object atributes (say, you pick the object color twice, but no shape). if this is excluded, please make it clearer.
- (5): this formulation seems to be the same/very similar (depending on $d$) to the InfoNCE objective. Please refer to that to establish the connection

### Minor points
- L91: spatially non-exclusive factors could be a bt misleading, maybe consider "global" (my guess based on the exampels you provide)
- L108: define $\theta, \phi$
- **Limitations:** though the product space assumption is prevalent in the compositionality literature, all compositions might not imply valid configurations (e.g., hallucinations, or placing anachronistic objects in an image, e.g. a television into a medieval scene). I'd add this to either the limitations or to L52
- L61: please reference the DCI paper
- (4) - why do you calll it $\mathcal{L}_{prior}?$ I believe it refers to the diffusion model, so maybe spell this out or call it the "diffusion loss" or something similar
- L220 (typo?): do you mean the image decoder? I don't understand how an encoder would take features
- Fig. 2: please add more information to the caption to make it self-contained (at least refer to your description in L255 and onward)
- Fig. 3: the image shows $\emptyset,$ but the caption shows $\phi$ (al;so the green color is hard to see)
- **Metrics**: could you please add a short description regarding how your metrics (FG-ARI mIoU, mBO, GRAM, etc) are defined? It's fine in the appendix
- Could you plase add emphasis (bold/underline + explanation) to Tab. 4?

---

> ### Author Rebuttal · Authors · 2025-07-31
>
> We thank to the reviewer for their time and effort in providing constructive review.
>
> > W1. Comparison to Wiedemer et al., 2024 [1]
>
> We appreciate the valuable comments. We agree that the distinction between our method and Wiedemer et al. [1] should be more clearly stated. While both approaches share the conceptual goal of enabling generalization to novel compositions of factors in disentangled representation learning, there are fundamental differences in how we achieve valid generalization.
>
> As the common part, both methods recognize that valid generalization requires two key components: first, compositions of disentangled representations must yield valid data (e.g., realistic images or representation $z^c \in Z$), and second, the composite representation $z^c$ must properly encode the information from its corresponding composite image $x^c$ to satisfy the representation learning objective. To address the second requirement, both our method and [1] introduce a compositional consistency loss with minor differences but identical naming. However, the approaches diverge significantly for the first component–how to ensure the validity of composite representations.
>
> Wiedemer et al. [1] employ an additive decoder to ensure valid compositions of disentangled representations (Sec. 3.1 in [1])). However, additive decoders are known to be unscalable for complex scenes, as their local decoding mechanism cannot capture complex interactions between objects due to limited expressive power [2]. More critically, additive decoders are designed based on the spatial exclusiveness bias (Def. 4 in [1]) , which assumes that each pixel should be affected by only a single latent variable. This limits their applicability to object-centric learning scenarios, preventing generalization to other disentanglement tasks.
>
> In contrast, our method ensures validity through a prior loss (Eq.4) that leverages SDS loss [2] to encourage both $z^c$ and its corresponding composite image $x^c$ to be realistic. As we do not require additive decoder anymore, our approach allows us to utilize an expressive diffusion decoder for modeling complex scenes. Crucially, since our method does not embed factor-specific biases like spatial exclusiveness into the architecture, it can be applied beyond object-centric learning to attribute disentanglement or joint disentanglement scenarios.
>
> More importantly, to the best of our knowledge, we are the first to demonstrate that different underlying factors of variation can be disentangled simply by adjusting the factor-specific mixing strategy, distinguishing our approach from previous methods that introduce factor-specific biases through architectures (e.g., spatial decoders in [1]) or loss functions (e.g., total correlation loss in FactorVAE).
>
> Finally, we provide a quantitative comparison of [1] with our method in attribute-, object-, and joint disentanglement. For a fair comparison, we used the same encoder for [1] as ours and only changed the decoder to an additive decoder. As expected, [1] cannot disentangle attribute factors at all (Shapes3D, CLEVR-Style), since they violate the spatial-exclusiveness assumptions. Moreover, in object disentanglement, we found that an additive decoder alone cannot disentangle objects (in fact, [1] validated their method only on very simple 2D synthetic datasets). When we additionally use the slot-attention module in [1], it reasonably disentangles objects in CLEVR dataset but still significantly inferior to our method in CLEVR-Style possibly due to the limited expressive power of the additive decoder. Additionally, we observed that object-wise manipulation with [1] always leads to unrealistic images with transparently overlapping objects due to the lack of interactions between latents inside the additive decoder.
>
> ||Shapes3D||CLEVR||||CLEVR-Style||||||
> |---|---|---|---|---|---|---|---|---|---|---|---|---|
> ||FactorVAE|DCI|Shape|Color|Material|Position ($\downarrow$)|Acc|GRAM ($\downarrow$)|Shape|Color|Material|Position ($\downarrow$)|
> | Ours | **0.975** | **0.837** | **87.04** | **93.93** | **94.81** | **0.032** | **96.5** | **5.05** | **83.56** | **90.48** | **93.74** | **0.053** |
> | Additive Decoder w/ SA | - | - | 33.86735 | 15.2355 | 51.8744 | 0.76527 | 23.3 | 18.585 | 34.20378 | 13.53733 | 50.89715 | 0.51651 |
> | Additive Decoder w/o SA | 0 | 0.031 | 82.90612 | 93.14322 | 91.78148 | 0.1095 | 23.3 | 15.84 | 35.96604 | 20.98686 | 54.10125 | 0.51978 |
>
> [1] Provable Compositional Generalization for Object-Centric Learning, ICLR24
> [2] Text-to-3D using 2D Diffusion, ICLR23
>
> > W2. Rephrasing the Abstract
>
> We thank the reviewer for the valuable comments.
> We will revise the Lines 8-16 to clarify the contribution of mixing strategies:
> “Our key insight is that different factors obey distinct “recombination rules” in the data distribution: global attributes are mutually exclusive (a face has one nose), whereas objects share a common support (any subset of objects can co‑exist). By training the model to maximize the likelihood and self‑consistency of images reconstructed from latents that are randomly remixed according to these rules, we force the encoder to discover whichever factor structure the mixing strategy reflects, without modifying the encoder, decoder, or loss terms."
>
> > W3. Better notation for objects and attributes
>
> Superscript and subscripts denote image index and latent index, respectively. Both an attribute and an object latent are denoted with a single latent vector $z_i$, where which types of factors to be encoded in latent vector $z_i$ is determined by our mixing strategies (Eq. 1,2,3). That being said, if latent vector $z_i$ is mixed with other latents following Eq. 1 or 2 when training, the resulting $z_i$ will capture attribute or object, respectively. This is core motivation of Eq.3 that induces the joint training of attribute and objects. First M latents $z_{1:M}$ and rest of the K-M latents $z_{K-M:K}$ are mixed with Eq.1 and Eq.2, respectively, promoting the learning of attributes on first M latents and objects on rest of them. We empirically found that first latent which is mixed with Eq.1 always captures style attributes, while the rest of the slots encodes object concepts in Sec. 4.3. We will clearly state it in the main paper.
>
> > W4 / W8. Reference for the InfoNCE and the DCI paper
>
> We will add references to the InfoNCE and DCI papers. For InfoNCE, we will further clarify its connection to our objective in the main paper.
>
> > W5. Misleading term in Line 91
>
> We use the term spatially non-exclusive to contrast with spatial exclusivity. We will revise the term to 'global' for clarity.
>
> > W6. Missing definitions for $\theta$ and $\phi$
>
> We will clearly define $\theta$ and $\phi$ as the learnable parameters of the encoder and decoder, respectively, in L108.
>
> > W7. Suggestions for adding limitation
>
> We will add the discussion in the limitation section.
>
> > W9. Why do you call it $\mathcal{L}_{prior}$?
>
> We followed [20] for coherent terminology. As this loss is not for training the diffusion model, but rather optimizing the given data with a pretrained diffusion model that estimates a data prior.
>
> > W10. Clarification on the image decoder (L220)
>
> We first encode the given image with pretrained vae model and we feed this vae latents into our encoder. It resembles Latent Diffusion model where input to the diffusion model is vae-features instead of RGB images. We will clearly state it in our main paper.
>
> > W11/W12. Suggestion for Fig.2 and Fig.3
>
> For Fig.2, we will incorporate more information from the main paper to improve the caption so that it is self-contained, and we will refer to the descriptions in L255 and onward. For Fig.3, we will revise the caption and color for better readability.
>
> > W13. Adding descriptions for metrics.
>
> Following the reviewer's suggestion, we will add descriptions regarding our evaluation metrics (FG-ARI, MBO, GRAM, DCI, etc.) in the revised version.
>
> > W14. Adding emphasis to Tab. 4
>
> We will add emphasis in our main paper.
>
> > Q1. Intuition for performance on Cars3D
>
> To clarify, we argue that our method is not worse on Cars3D because it significantly outperforms the baselines on the DCI metric, despite being inferior on FactorVAE score. The major drop in score in our method is mainly due to the disentanglement of the rotation factor into horizontal axis and angle (Fig. 9 in Appendix), which might be penalized when a single factor is not correctly aligned with the GT factor (rotation). In contrast, the major performance of DisDiff seems to come from the entanglement of factors such as car shape and color, as shown in the last row of Fig.5 in the DisDiff paper.
>
> > Q2. Pseudocode for matching technique (A.7)
>
> Here is the pseudo code for our matching technique to identify object region:
>
> ```
> x : an image
> z : Object representation of x, i.e., z = enc(x)
> n : the number of latent vectors
> x_ref : randomly sampled reference B images
> def matching(z, x, x_ref):
>     # Generate mixed latent representations
>     z_ref = [encode(_x) for _x in x_ref]
>     z_mixed = [replace_ith_latent(z_ref, z, i) for i in range(n)]
>     x_mixed = [decode(_z) for _z in z_mixed]
>
>     # cm of imgs
>     x_ref_cm = x_ref.mean(dim=0)
>     x_mixed_cm = x_mixed.mean(dim=1)
>
>     # Metric 1
>     s = softmax([1 - distance(x, x_mixed_cm).mean(-1)])
>
>     # Metric 2
>     d = softmax([distance(x_mixed_cm, x_ref_cm)].mean(-1))
>
>     return (s + d).argmax(dim=0) # mask
> ```
>
> > Q3. Baseline models hyperparameter tuning
>
> For the attribute disentanglement experiment, we report the values from the DisDiff paper, so we do not train baseline models ourselves.
>
> For object disentanglement and joint disentanglement, we aligned the hyperparameters (e.g., depth) for model architectures of LSD and L2C with ours. For all experiments, we used the best-performing learning rate, which turned out to be similar across all baselines and our methods.

---

> > ### Comment · Reviewer_XmPJ · 2025-08-04
> >
> > Thank you for your clarificiations!
> >
> > All but one of my concerns is resolved now, the one remaining is regarding [1]. I do acknowledge that your work provides empirical contributions, but what still concerns me is that, based on your response, it remains potentially unclear to the reader that the consistency loss is not a novel contribution. It should be made clear in the storyline that you are heavily building on prior works in compositionality, and your main contribution is proposing mythological improvements that provide compositionality in scenarios going beyond prior works while using more flexible models.

---

> ### Author Response · Authors · 2025-08-06
> **Reply to Reviewer XmPJ**
>
> We appreciate the reviewer's detailed feedback. We agree with the reviewer that our presentation of the consistency loss could potentially be misleading to readers, although we did not claim it as our novel contribution.
>
> For clarity, we will first clearly state that the consistency loss follows prior work [1] in compositionality in Sec. 3.3 of our main paper. Specifically, we will revise Lines 188-191 as follows:
>
> “Minimizing the prior loss alone could lead to a degenerate solution such as generating arbitrary realistic composite images $x_c$ regardless of the given $z_c$. To prevent such degeneracy, we adopt compositional consistency loss from [1] to ensure alignment between $z_c$ and the inverted latent ${\hat z_c} = E_{\theta}(D_{\phi}({z}_c))$ so that ${x}_c$ should be faithfully reflect the contents of ${z}_c$. However, empirical observations reveal that direct minimization of absolute distance is insufficient to prevent misalignment between ${x}_c$ and ${z}_c$. In practice, … (further explanation on how we reformulate consistency loss to Eq. (5)).”
>
> Then, we will clearly highlight our core contribution as "designing a framework that can disentangle different underlying factors of variation (including attributes and objects) simply by adjusting the factor-specific mixing strategy, distinguishing our approach from previous methods that introduce factor-specific biases through architectures and loss functions" throughout the paper. Additionally, for a clear comparison to [1], we will provide empirical studies (as in our previous response) and in-depth discussion to highlight the difference of our work from [1].
>
> We hope that these clarifications and adjustments will adequately address the concerns raised, and we would appreciate any further feedback regarding this point.
>
> [1]  Provable Compositional Generalization for Object-Centric Learning, ICLR24 [2] Text-to-3D using 2D Diffusion, ICLR23

---

> > ### Comment · Reviewer_XmPJ · 2025-08-07
> >
> > Thank you for your response!
> >
> > I like the proposed addition to the main text, and as I said before, your paper has a lot of value in advancing compositional generalization from a methodological point of view.
> > To make clear for all readers what your exact contribution is, **please specify your contributions in the abstract as well** (i.e., that you borrow the consistency loss from prior works) **and post your proposed new abstract on openreview.**
> > Again, this is only to ensure that your contributions (which, I believe, are worthy of acceptance) will be clear to all readers. If you post an abstract like that, I will be raising my score and will be advocating for acceptance.

---

> > > ### Author Response · Authors · 2025-08-08
> > > **Reply to  Reviewer XmPJ**
> > >
> > > We appreciate the thoughtful comment. Following the reviewer’s suggestion, we have revised the abstract to explicitly state that we adopt the compositional consistency loss from Wiedemer et al. (2024), and post our new abstract below:
> > >
> > > **Modified version of Abstract**
> > >
> > > Recent disentangled representation learning (DRL) methods heavily rely on factor-specific strategies—either learning objectives for attributes or model architectures for objects—to embed inductive biases. Such divergent approaches lead to significant overhead when novel factors of variation do not align with prior assumptions, e.g., statistical independence or spatial exclusivity, or when multiple factors coexist, as practitioners must redesign architectures or objectives. To address this, we propose a compositional bias, a modular inductive bias decoupled from both objectives and architectures. Our key insight is that different factors obey distinct "recombination rules" in the data distribution: global attributes are mutually exclusive (a face has one nose), whereas objects share a common support (any subset of objects can co‑exist). We therefore randomly remix latents according to factor‑specific rules, i.e., mixing strategy, and force the encoder to discover whichever factor structure the mixing strategy reflects through two complementary objectives: (i) a prior loss that ensures every remix decodes into a realistic image, and (ii) the compositional consistency loss introduced by Wiedemer et al. (2024), which aligns each composite image with its composite latent. Under this general framework, simply adjusting the mixing strategy enables disentanglement of attributes, objects and even both without modifying the objectives or architectures. Extensive experiments demonstrate that our method shows competitive performance in both attribute and object disentanglement, and uniquely achieves joint disentanglement of global style and objects.
> > >
> > > We hope this revision makes our contribution clear. If any further adjustments to the abstract or wording would be helpful, we are happy to refine it.

---

> > > > ### Comment · Reviewer_XmPJ · 2025-08-08
> > > >
> > > > Thank you, this addresses all of my concerns, I raised my score to 5.

---

### Official Review · Reviewer_1m9R · 2025-07-01

**Clarity:** 3
**Significance:** 3
**Originality:** 4
**Rating:** 5
**Confidence:** 4

**Summary:**

This paper learns disentangled representations of attributes and objects through self-supervised methods, which is a highly meaningful and interesting research direction. By designing different mixing strategies and learning strategies, the method achieves highly effective spontaneous decoupling, and demonstrates relatively obvious advantages over existing methods in multiple metrics across various datasets. Previous Disentangled Representation Learning was mainly based on the Slot Attention framework. This work designs new learning objectives and architectures, achieving better performance in some aspects, which I consider highly innovative and appealing.

**Questions:**

Question: I would like to ask whether an object is represented by a single latent or multiple latents. My understanding is that the dimensions of the representation are grouped, and an object has multiple attribute representations. I wonder if my understanding is correct. If so, does it mean that specific channels of the representation encode specific attributes?

**Ethical Concerns:**

["NO or VERY MINOR ethics concerns only"]

**Final Justification:**

This is an interesting work with good presentation. The authors' responses have well addressed my questions. I believe it meets the standards for publication and will maintain my rating.

**Limitations:**

Yes

**Paper Formatting Concerns:**

No concerns.

**Quality:**

4

**Strengths And Weaknesses:**

Strengths:
1. Propose attribute mixing strategy and object mixing strategy, which is the key to achieve attribute and object disentangle.

2. Propose novel learning objectives and the experimental results show its effectiveness.

3. Paper is well-organized, and the experiments are comprehensive.


Weeknesses:
1. Objective function is relatively complex, which may limit its potential and scaling to more data.

---

> ### Author Rebuttal · Authors · 2025-07-31
>
> We thank to the reviewer for their time and effort in providing constructive review.
>
> > W1. Objective function is relatively complex, which may limit its potential and scaling to more data.
>
> We thank the reviewer for the valuable comment. Although our objective function consists of several components, we empirically observe that our method works well across three different tasks (object-, attribute-, joint-disentanglement) and seven different benchmarks, which implies the robustness of our objective functions. Moreover, our ablation study (Table 5), reveals that all objective functions are crucial for our method.
>
> Still, we agree that multiple objective functions of our method may require tuning of hyperparameters in larger and complex data. Improving our work into a more generalized version and stabilizing it for larger, complex data will be our important future work.
>
>
> > Q1. Whether an object is represented by a single latent or multiple latents.
>
> In our implementation, both attributes and objects are represented by vector representations. Although objects are represented by multiple dimensions, each dimension does not have to encode specific attributes unless we impose additional constraints. As the reviewer suggested, objects could be regarded as a group of attributes so we would be able to extend our mixing strategies to disentangle those factors with hierarchy, i.e., scenes are decomposed into objects, and objects are further decomposed into attributes. We believe this is an interesting and important direction for our future work.

---

> > ### Comment · Reviewer_1m9R · 2025-08-05
> > **Maintaining my original rating**
> >
> > Thank you for your reply, which has addressed my question. I consider this a interesting work, and I tend to maintain my original rating.

---

### Official Review · Reviewer_Yeku · 2025-07-02

**Clarity:** 3
**Significance:** 1
**Originality:** 2
**Rating:** 4
**Confidence:** 4

**Summary:**

The paper proposes a disentanglement framework that decouples factor-specific inductive biases from learning objectives and architectures, enabling both attribute and object disentanglement under a single set of objectives and architectures. The framework derives mixing strategies as a compositional bias, embedding factor-specific biases in a modular way, and proposes specific strategies for attribute, object, and joint disentanglement.

**Questions:**

According to the weaknesses I previously highlighted, addressing the following points could potentially change my opinion:
	A thorough discussion clarifying the essential differences between the proposed “mixing strategy” and the commutative principle or Group-based DRL training methods.
	More convincing visual comparisons with baseline models to substantiate the claims.
	Experimental results conducted on real-world multi-object datasets to demonstrate practical applicability.

**Ethical Concerns:**

["NO or VERY MINOR ethics concerns only"]

**Final Justification:**

Sorry for the delay due to some personal issues, and I have already read all comments and discussions about this paper and have finally decided to raise my initial score to 4.

**Limitations:**

Yes

**Quality:**

2

**Strengths And Weaknesses:**

Strengths
	The concept of separating attributes and object disentanglement is straightforward but significant for the DRL domain.
	The Mixing Strategy is clearly articulated, with well-defined designs for losses and objectives.
Weaknesses
	Training a DRL model using the commutative principle within Group theory is a common approach. This method exchanges specific factors between two different images or objects for reconstruction, encouraging the model to learn the attributes of each specific factor. The authors are encouraged to clearly highlight the differences between this work and existing Group-based DRL methods to demonstrate the novelty of their approach.
	In lines 90-93, the authors claim distinctions between their object disentanglement strategy and other object-centric DRL models. To support these claims, more visual qualitative comparisons with state-of-the-art object-centric DRL models should be provided in Section 4.
	Although quantitative results on attribute disentanglement are presented in Table 1, a visual comparison with baseline methods is still necessary to validate the superior performance of the proposed model. This could be included as an extension of Figure 2, showcasing comparisons with baselines.
	The experiments are currently conducted only on toy datasets such as CLEVR and Shape3D, which limits the contribution of this work to the DRL community. Given that the study aims to disentangle attributes and objects simultaneously, the authors should consider evaluating their method on real-world multi-object datasets to demonstrate its practicality.

---

> ### Author Rebuttal · Authors · 2025-07-31
>
> We thank to the reviewer for their time and effort in providing constructive review.
>
> > W1/Q1. Comparison to existing Group-based DRL
>
> As the reviewer pointed out, disentangled representation learning using Group theory is indeed an actively researched area and related to our work. Group theory-based DRL typically define disentangled representation $Z$ as follows: Given ground truth factors of variation $W$ and decomposable group $G$ (i.e., $G=G_1\times G_2\times …\times G_n$), the representation $Z$ is disentangled w.r.t. $G$ if (1) there exists a mapping $f$ from $W$ to $Z$ such that $f(g\cdot W)=g\cdot f(W)$ for all $g\in G$ and $w\in W$, and (2) there is a decomposition $Z=Z_1\times…\times Z_n$ such that each $Z_i$ is affected only by $G_i$.
>
> Despite this convincing principled definition, since the GT factors of variation $W$ are infeasible to obtain in unsupervised DRL, existing unsupervised methods often utilize necessary conditions for group actions of $G$ applied to $Z$ and disentangled representation $Z$ [1,2]. For instance, [1], the related work closest to our method, defines permutation group actions of element-wise addition on $Z$ and introduces losses to enforce commutativity and cyclicity of group actions.
>
> Our method takes a similar approach, but with important distinctions. From a Group theory perspective, unlike existing work, the group action in our method is defined on disentangled representation pairs $(z^1, z^2)$ rather than a single latent $z_i$. By defining the group action on a pair $(z^1, z^2)$, we can impose additional necessary conditions for how each underlying factor combines to generate observations, which cannot be induced by group action defined on a single latent $z_i$. For example, commutativity and cyclicity of group action are necessary conditions for both attributes and objects but do not impose attribute or object-specific properties.
>
> Our main contribution here is that we define group action as factor-specific mixings (i.e., permutations) between two latents and demonstrate that this additional necessary condition imposes effective factor-specific inductive bias for attribute and object disentanglement without changing the overall learning objectives or model architectures. To the best of our knowledge, we are the first to study differently learned disentangled representations of different factors of variation through a mixing strategy (or a form of group action) without employing factor-specific architectures or learning objectives.
>
> [1] Towards building a group-based unsupervised representation disentanglement framework, ICLR22 \
> [2] Commutative lie group vae for disentanglement learning, ICML21
>
>
> > W2/Q2. More visual qualitative comparisons with SoTA object-centric DRL methods
>
> We thank the reviewer for the valuable comments. Due to the policy that prohibits providing any links during rebuttal, we cannot provide qualitative results during rebuttal.
> Instead, we describe the differences in qualitative results here and will provide visual qualitative comparisons to SOTA object-centric methods in the revised version of the paper.
>
>
> In Lines 90-93, we claim that existing slot attention-based methods fail to disentangle spatially non-exclusive attributes such as global styles. In joint disentanglement experiment (Sec. 4.3), we observed that both LSD [3] and L2C [4] fail to disentangle style information into a single latent representation. When we decode all K × K possible latent exchanges between two images to identify the latent that encodes the style information, we found that exchange of single latent does not alter the global style, implying that style information is distributed along multiple latent representations. Therefore, the accuracy for both LSD and L2C are very low in Table 4. In contrast, our model successfully alters the global style of the scene without affecting the objects, which shows that ours successfully captures the style information into a single latent.
>
> [3] Object-Centric Slot Diffusion, NeurIPS23 \
> [4] Learning to Compose: Improving Object Centric Learning by Injecting Compositionality, ICLR24
>
>
>
> > W3/Q2. Qualitative comparison of attribute-DRL with baselines
>
> Again, due to the policy that prohibits providing any links during rebuttal, we will provide a description of qualitative results and further provide them in the revised version later.
>
>
> For qualitative comparison, we choose the strongest baseline, DisDiff [5] and perform attribute-wise swapping as in Figure 2.
> For the Car3D dataset, DisDiff often produces unrealistic samples as shown in the second column of Figure 5 in the DisDiff paper. While DisDiff shows reasonable manipulation along each attribute, we found that shapes are often entangled with car colors.
> Moreover, for the MPI3d dataset, we found that changing a single attribute, e.g., object shape in DisDiff often changes other attributes such as object colors or sizes.
>
> In contrast, our method generally shows better disentanglement in terms of attribute swapping except in the extreme cases when the size of the object is too small so that it is hard to distinguish the attributes in the image. We found that both disdiff and our method perform well on the Shapes3D dataset.
>
> [5] Unsupervised Disentanglement of Diffusion Probabilistic Models, NeurIPS23
>
>
> > W4/Q3. Evaluation on real-world multi-object datasets
>
> To address the reviewer's concern, we conducted additional experiments on the more realistic multi-object dataset, MultiShapeNet (MSN) dataset (Please refer to bottom rows in Table 3 of [6]). It includes 11733 unique shapes of furniture with increased overall complexity of shapes compared to CLEVR datasets. We compare our method with object-centric approaches using unsupervised segmentation.
>
> The model architecture and hyperparameters were kept the same as in the previous object disentanglement experiments. We measure FG-ARI, mIoU, mBO on object masks following common practices in object-centric literature. As our method does not have a built-in mechanism to directly express group memberships between pixels, we additionally train a Spatial Broadcast Decoder on the frozen latent representations to predict explicit object masks for each latent representation as described in Section 4.2. The results are reported in the below table.
>
>
> | Model | FG-ARI | mIoU | mBO |
> |---|---|---|---|
> | SLATE | 70.44 | 15.55 | 15.64 |
> | LSD | 67.22 | 15.39 | 15.46 |
> | Ours | **76.34** | **36.69** | **36.90** |
>
> Among the slot-attention based baselines, our method achieves the best performance across all of three metrics, even though ours does not employ any spatial clustering mechanism like slot attention. These results demonstrate the effectiveness of our framework in disentangling object representations in a complex dataset.
>
>
> [6] Stelzner el al., Decomposing 3D Scenes into Objects via Unsupervised Volume Segmentation, in ICLR 22.

---

> > ### Author Response · Authors · 2025-08-08
> > **Reply to Reviewer Yeku**
> >
> > Dear Reviewer Yeku,
> >
> > We would like to sincerely thank the reviewer again for their valuable comments. We kindly follow up to inquire whether our response has sufficiently addressed the reviewer’s questions. Should the reviewer have any further queries or concerns, we would be happy to discuss.

---

### Official Review · Reviewer_Z8su · 2025-07-02

**Clarity:** 3
**Significance:** 2
**Originality:** 2
**Rating:** 5
**Confidence:** 3

**Summary:**

This paper is about disentangling attributes and objects in an image with a common framework. The idea is to optimize image reconstructions by mixing factors in the latent. The approach is assessed on public datasets and compared with existing approaches.

**Questions:**

In addition to or as a complement to what was reported in the weaknesses section, I would like the authors to clarify the following points:
- How does the field of object-centric learning relate to object disentanglement? Somehow, they seem to be used as synonyms
- How does the proposed method compare to weakly-supervised learning approaches? Or, in alternative, why have such methods not been considered?
- The differences with [20] (ref. in the main paper) should be more clearly stated. For instance, they also use an encoding-decoding (a UNet)
- Rows. 137-139: I find this example misleading. Isn't this more related to object disentanglement, since it reasons on object parts?
- Row 162: Is the separation of attribute types in the latent forced at training time? In general, some more details on the training would be appreciated
- Row 186: Is the final targeted downstream task image generation?
- Eq. (5); Shouldn't we have an average on the bottom part of the fraction? As it is now, my impression is that the numerator and denominator are not numerically comparable
- Sec. 4.1 and 4.2: I am not sure what version of the method we are talking about here. For instance, are we evaluating in Sec. 4.1 the relevant portion of the latent (i.e. the first M attributes) jointly learnt with the rest, or is it a "partial" learning of only that part?
- As already mentioned, although what is presented in the experiments on joint disentanglement is nice, the fact that only one dataset and only one style attribute are used should be justified, as it looks like a limitation

**Ethical Concerns:**

["NO or VERY MINOR ethics concerns only"]

**Final Justification:**

Considering the reviews, the rebuttal and the discussion with the authors, I confirm my positive opinion on this paper. In the rebuttal, the authors successfully addressed my doubts, in particular they provided further experimental evidence of the joint disentanglement, which is a key contribution of this work.

**Limitations:**

It is mentioned in Sec. 5 that limitations are discussed in the Appendix. However, there is no Appendix to the paper, while the information can be found in the Supplementary Material.

**Paper Formatting Concerns:**

The paper looks correctly formatted

**Quality:**

2

**Strengths And Weaknesses:**

This paper represents a nice reading, overall very clear (with few exceptions). The proposed idea shows a good level of novelty, and it may have a potential impact on different tasks where managing interpretable representations can be an add-on.

On the weaknesses, I want to mention the following points:
- The introduction is not fully clear, and it fails to appropriately prepare the reader for the rest. Maybe, examples to give the intuitions behind attribute and object disentanglement could help to shape the story
- On the related works, in my understanding, the class of disentanglement methods relying on weak-supervised (e.g. [1]) is not discussed. However, such methods may have the potential to address the challenges in this paper and might be of interest.
- Again, on the related works, it would be useful to clarify the difference between the concepts of "object disentanglement" and "object-centric learning". To the unfamiliar reader, the impression is that the terms are sometimes used as synonyms
- On the training, I am not sure I fully got the details. For instance, learning the disentanglement model adopted later for mixing is a step included in the training of the proposed approach, or is the idea to rely on pre-trained models?
- The experiments are nice, but only refer to synthetic data, while in more general (possibly real) scenarios, conditions are more challenging  (for instance, we may have FoVs distributed on more than one component of the latent, or mixed). Some clarifications on the generalisation would help
- Again, on the experiments, I think that weakly-supervised methods should be taken into account
- Regarding the joint disentanglement, I have a couple of doubts. The first one is that while this is the core contribution of the paper, it is evaluated on one dataset only, and this is somehow a limitation. The second doubt refers to the fact that, as far as I understand, only one style attribute is considered, against multiple object-based properties. This leads me to another doubt that I have: in Shapes3D, why are we not considering floor and wall colours as style attributes? A clarification would help appreciate the difference
- At different points, it is stated that the method can freely choose model architectures. If I'm corretly interpreting the statement, this would require showing experiments involving at least another architecture, to show this ability


[1] https://arxiv.org/pdf/2002.02886

---

> ### Author Rebuttal · Authors · 2025-07-31
>
> We thank to the reviewer for their time and effort in providing constructive review.
>
> > W1. Clarification on the Introduction.
>
> Following the reviewer’s suggestion, we provide
> a high-level intuition for how we derive compositional bias to disentangle attributes and objects as follows:
>
> Attributes and objects represent two distinct types of generative factors that differ in how they combine to form valid scenes (see Fig.1). Each attribute factor appears exactly once in a scene (e.g., a face has only one nose), whereas multiple instances of object factors can appear independently (e.g., varying numbers of cars on a road).
>
> Inspired by this, we formulate those ‘valid’ compositions as mixing strategies and achieve disentanglement by ensuring that composite images generated through these mixing strategies appear realistic.
>
>
> > W2/W6/Q2. Comparison to weakly-supervised method.
>
> We thank for pointing out the relevance of weakly-supervised (WS) disentanglement methods.
>
> Indeed, there is a line of work that addresses disentangled representation learning through weak supervision [1, 2]. A typical approach utilizes observation pairs that share a certain number of factors of variation (FoV). For instance, [2] assumes we are given paired observations for which some (but not all) FoV have the same values. Such additional knowledge from weak supervision has shown effectiveness in identifying the underlying Fov.
>
> Meanwhile, WS methods are not always useful, since such observation pairs are often hard to obtain, e.g., finding two people with identical eyes or nose. Thus, significant research efforts are focusing on the more challenging unsupervised setting of disentangled representation learning, which is our main scope. Note that, as [3] propose that purely unsupervised learning of disentangled representations is infeasible, many recent works design and utilize inductive biases for each underlying factor of variation(FoVs). While these inductive biases are mostly factor-specific and non-compatible with each other, our main contribution is a common framework that can handle various FoVs.
>
> Nevertheless, the relationship between our method and WS approaches and the application of our method to the WS setting seems to be an interesting direction. In the current version, our method is not directly compatible with WS methods, since objective functions of those methods are mostly restricted to VAE formulations. Thus, designing a more general approach, in terms of model or loss, for the WS setting is needed. However, we think this is out of scope in the current paper and leave it as future work.
>
> [1] Weakly Supervised Disentanglement with Guarantees, ICLR20 \
> [2] Weakly-Supervised Disentanglement Without Compromises, ICML20 \
> [3] Common assumptions in the unsupervised learning of disentangled representations, ICML19
>
>
> > W3/Q1. Clarification on "object-disentanglement" and "object-centric learning".
>
> We used those terms as synonyms. We will unify the terminology as 'object-centric learning' for clarity.
>
> > W4. Clarification on training.
>
> The disentanglement model (encoder) is jointly trained with the reconstruction and compositional consistency objectives computed from composite images.
>
> > W5. More general scenarios
>
> We agree with the reviewer that the datasets used handle only simple scenarios with statistically independent factors of variation (FoVs), unlike real-world cases.
>
> To verify our method on more challenging scenarios with correlated factors, we followed [4] to generate correlated benchmarks. We introduced 0.1 correlation between 1/2/3 pairs of GT factors in Shapes3D, as in [4], then trained and evaluated our models using FactorVAE and DCI metrics across 3 seeds (see table below).
>
> |Amount of Correlation|FactorVAE|DCI|
> |:---:|:---:|:---:|
> |No Correlation|0.975±0.059|0.837±0.105|
> |Pairs:1, $\sigma=0.1$|0.930±0.061|0.798±0.082|
> |Pairs:2, $\sigma=0.1$|0.997±0.002|0.806±0.060|
> |Pairs:3, $\sigma=0.1$|0.965±0.030|0.801±0.024|
>
> Our method showed only slight metric drops despite correlations. Unlike prior work that relies on statistical independence between latent variables (e.g., Total Correlation), our compositional objective only encourages discovering atomic factors whose compositions are valid under predefined mixing strategies. This allows successful disentanglement even with correlated factors, as the compositional requirement doesn't penalize correlations.
>
> [4] Disentanglement of correlated factors via hausdorff factorized support, NeurIPS23.
>
>
> > W7/Q9. Regarding the joint disentanglement.
>
> Due to the limited time during the rebuttal period, our additional experiment on joint disentanglement is not finished yet. We will provide a joint disentanglement experiment on MultiShapeNet dataset, which has increased complexity of objects with various furnitures, and results will be provided in the second discussion round.
>
> As stated in our manuscript, we refer to attribute factors as globally-shared scene properties, e.g., color, style, whereas object factors are distinct spatial entities, such as individual objects. In Shapes3D, all of the data consist of unique floor and wall colors, so we classify them as attributes.
>
> > W8. Clarification on model architectures.
>
> Lines 253-254 state our method can "freely choose model architectures.." but this doesn't mean it works well with arbitrary architectures.
>
> To elaborate, while it has been shown that expressive power of the model architecture often becomes a bottleneck for disentanglement in complex scenes [6,7,8], conventional unsupervised disentangled representation learning methods are forced to use less expressive model architectures. Specifically, the VAE-based total correlation minimization loss [9,10] requires, by definition, factors to be encoded in single dimensions, making it difficult to use for vector-wise disentanglement. Additionally, object-centric learning methods [11] often rely on additive decoders to prevent interaction between latent variables. Those architectures work adequately on relatively simple synthetic datasets but such independent modeling cannot handle complex scenes involving interaction between objects [7]. These constraints enforce model architectures that often lead to lower performance, i.e., dimension-wise encoding or less expressive decoders. In contrast, our method is designed to be free from such architectural constraints that can bottleneck performance, allowing for sufficient expressive power, e.g., with vector-wise encoding and powerful diffusion decoder.
>
> The intention in our paper was to use the term 'freely' in this context. However, we agree that the current writing could be misleading, and will clarify it in the revision.
>
> [6] Unsupervised Disentanglement of Diffusion Probabilistic Models, NeurIPS23 \
> [7] Illiterate DALL-E Learns to Compose, ICLR22 \
> [8] Object-Centric Slot Diffusion, NeurIPS22 \
> [9] Disentangling by Factorising, ICML18 \
> [10] Isolating Sources of Disentanglement in VAEs, NeurIPS18 \
> [11] Provably Compositional Generalization, ICLR24
>
>
> > Q3. The differences from [20]
>
> As stated in Lines 117-122, the main difference from our work is that [20] heavily relies on object-specific architecture such as slot-attention (SA) encoder and thereby cannot disentangle spatially non-exclusive factors, e.g., attributes. In contrast, our method does not impose factor-specific bias on the model architecture enabling disentangle both attributes and objects on the same model architectures (Sec 4.3).
>
> Specifically, in terms of model architecture, [20] employs an auto-encoding framework where the encoder includes a SA module. In contrast, our method replaces the SA in QFormer to avoid spatial-exclusiveness biases within SA (Line 222), enabling application into attribute disentanglement.
>
> In terms of learning objective, [20] uses unconditional diffusion model (DM) for maximizing the likelihood of the composite image, while we propose to use conditional DM for likelihood and modify the objective accordingly, as score estimation of unconditional DM is more robust to OOD compared to that of conditional DM.
>
> Please refer to Lines 179-187 and Appendix A.9 for detailed discussion. We will revise our paper to more clearly state these distinctions.
>
>
> > Q4. About example in L137-139
>
> We used eyes, a nose, a mouth within a face as an intuitive example for attributes, since those factors are uniquely determined in each face. However, we agree with the reviewer that it could be misleading as those components can be spatially decomposed. We will revise the example into spatially non-exclusive ones such as color, texture, material of the ball to clearly distinguish it from object factors.
>
> > Q5. The separation of attribute types at training
>
> Yes, they are separated in the training stage. Given a pair of two images in the training stage, first M latent representations for each image are mixed with attribute mixing strategies (Eq.1) while the rest of K-M  latents are mixed with object mixing strategies (Eq.2). Then this composite representation is decoded into image via Decoder $\phi$ and optimized with reconstruction, compositional consistency  and prior losses. We will clearly state details in the revised version of the paper.
>
> > Q6. The targeted downstream task
>
> In the context of Lines 186, image generation is not our downstream task but part of our training objective. Our encoder is optimized to learn latent representations such that composite images generated by predefined mixing strategies to be valid.
>
> > Q7. About Eq.5
>
> Note that taking the average of the denominator will not affect the overall training as 1/N (averaging) will be decomposed into a constant w.r.t. $\theta$.
>
> > Q8. Clarification on Sec 4.1 and 4.2
>
> We used Eq.1 to randomly mix the entire set of latents for Sec. 4.1. Similarly, we use Eq.2 for Sec. 4.2. Only Sec. 4.3 uses both of the equations (Eq.1 for first M and Eq.2 for K-M latents) and all of the latents are jointly optimized.

---

> > ### Comment · Reviewer_Z8su · 2025-08-04
> > **Thanks for the rebuttal**
> >
> > I wish to thank the authors for their detailed rebuttal, which clarified almost all the points I raised.
> > I have a couple of remaining doubts. The first one refers to additional experiments on joint disentanglement (currently ongoing), to get a better feeling of the stability and robustness of the proposed approach.
> > The second clarification I would ask is about the targeted downstream task. If it's not the generation (which is indeed part of the training, as stated by the authors in their rebuttal), can the authors clarify what may be the final aim?

---

> ### Author Response · Authors · 2025-08-07
> **Reply to Reviewer Z8su**
>
> We appreciate the thoughtful comments.
>
> > Q1. Additional results on joint disentanglement
>
> For additional experiments on joint disentanglement, we augment the MultiShapeNet (MSN) dataset with 4 different global styles as in our previous CLEVR-Style experiments. MSN includes 11,733 unique furniture shapes with increased complexity compared to CLEVR (Please refer to the bottom rows in Table 3 of [1] to see the MSN data). We compare our method with the strongest attribute-disentanglement baseline (DisDiff) and object-disentanglement baselines (LSD, L2C), reporting quantitative results in the table below.
>
> For style disentanglement, our method achieved the highest accuracy and the lowest GRAM loss, demonstrating that it successfully isolates style information in a single latent and transfers it faithfully to the target image. Since baselines lack an internal mechanism to specify the latent representation for style information, we decode all $K \times K$ possible latent exchanges between two images and choose the pair yielding the lowest GRAM loss. Even with their best‑case result (indicated by '*' in the table), they fell short of our performance, confirming that they do not reliably capture a style factor.
>
> For object disentanglement, our method produced the highest mIoU&mBO, indicating accurate localisation and tight boundaries for each object mask, while FG‑ARI was a bit lower than that of LSD and L2C. This trade‑off arises because our broadcast decoder generated tight, object‑internal masks. FG‑ARI—which measures membership between two pixels within an same group—penalizes such masks when GT segmentations are looser, whereas mIoU and mBO reward the improved boundary precision. In contrast, slot attention modules in LSD and L2C often generate larger masks that bleed into the background or even include other objects, which leads to higher FG‑ARI at the expense of mIoU&mBO. DisDiff, whose information‑theoretic objective is not designed for a varying numbers of objects, fails to effectively separate either style or object factors.
>
> For more clear inspection, we will add qualitative results showing style/object swapping (as in Fig. 5, 6 of our paper) in the paper.
>
> ||Acc($\uparrow$)|GRAM($\downarrow$)|FG-ARI($\uparrow$)|mIoU ($\uparrow$)|mBO ($\uparrow$)|
> |---|:---:|:---:|:---:|:---:|:---:|
> |DisDiff|37.0|10.8|5.62|7.69|9.13|
> |LSD|12.5|11.3|52.7|23.3|23.9|
> |L2C|81.0|8.35|**53.3**|28.4|28.8|
> |Ours|**100.0**|**6.59**|42.3|**44.1**|**44.2**|
>
>
> > Q2. Can the authors clarify what may be the final aim?
>
> We would like to clarify that disentangled representation learning (DRL) generally aims to **train an encoder capable of extracting independent latent factors** from a given image. Disentangled representations are widely recognized for their usefulness in several downstream tasks, such as: (1) systematic generalization to unseen combinations of latent factors, as required in compositional zero‑shot learning and symbol‑to‑image recombination benchmarks [2, 3]; (2) controllable image manipulation, in which individual latent factors are manipulated to change appearance while leaving all other content unchanged [4–6]; and (3) causal reasoning, in which separate latent factors can be individually intervened upon to support counterfactual analysis of their isolated effects [7]. Consequently, the DRL literature primarily focuses on developing robust algorithms that encourage an **encoder** to identify and separate latent factors, rather than tailoring the model to specific downstream tasks.
>
> Our work follows this general objective of the DRL field, and therefore we primarily evaluated the encoder's disentanglement capability on standard benchmarks using: (1) DCI or FactorVAE metrics for attribute disentanglement [4] and (2) object property prediction tasks for object-centric learning [5,6]. Additionally, we include attribute/object swapping results (where images are generated by swapping disentangled representations) as qualitative evidence that our encoder successfully captures and separates distinct attributes or object information, following previous work [4,5,6].
>
> While we acknowledge the importance of studying how such disentangled representations can further enhance specific downstream tasks, including image generation tasks, detailed exploration of those directions is beyond the scope of this paper.
>
>
> [1] Stelzner el al., Decomposing 3D Scenes into Objects via Unsupervised Volume Segmentation, ICLR, 22
>
> [2] Xin et al., Disentangled Representation Learning, TPAMI, 24
>
> [3] Montero et al., The Role of Disentanglement in Generalization, ICLR, 2021
>
> [4] Yang et al., DisDiff: Unsupervised Disentanglement of Diffusion Probabilistic Models,” NeurIPS, 23
>
> [5] Jiang et al., Object‑Centric Slot Diffusion,” NeurIPS, 23
>
> [6] Jung et al., Learning to Compose: Improving Object‑Centric Learning by Injecting Compositionality,” ICLR, 24
>
> [7] Schölkopf et al., Toward Causal Representation Learning, Proceedings of the IEEE, 21

---

> > ### Comment · Reviewer_Z8su · 2025-08-08
> >
> > Dear Authors
> >
> > Thank you for the detailed answer, which addresses my last questions. I confirm my positive rating on this pape. I would suggest you add some comments about the downstream task. I understand here the focus is on the disentangled model, and I'm aware of the potential of such models for different tasks. At the same time, mentioning possible tasks of interest for you (possibly the object of future works) may be useful to place the work in the framework/context foreseen by the authors.
> >
> > Rev. Z8su

---

> > > ### Author Response · Authors · 2025-08-09
> > > **Reply to Reviewer Z8su**
> > >
> > > We are pleased that our response addressed the reviewer's concerns and we sincerely appreciate the thoughtful feedbacks. We believe our method is applicable to downstream tasks such as controllable image manipulation (e.g., attribute‑ or object‑level edits) and object‑centric world‑model training that leverages our object‑disentangled representations, as exemplified by [1]. As suggested by the reviewer, we will add a brief discussion on possible downstream tasks for our work in the revised paper.
> > >
> > > [1] Jeong et al., Object-Centric World Model for Language-Guided Manipulation, ICLR 2025 Workshop on World Models.

---

### Note · Authors · 2025-08-13

We thank the reviewers and AC for their thorough evaluation and constructive feedbacks.

We appreciate the positive assessments from the reviewers:

* Well‑written, organized, and clear (Z8su,1m9R,XmPJ)
* Proposed idea shows novelty (Z8su)
* Clearly articulated mixing strategies with well-defined objectives (Yeku)
* Extensive experiments show effectiveness of our method (1m9R,XmPJ)

The main concerns include:
* Clarifying differences from prior work, especially [1] (XmPJ), weakly supervised methods (Z8su), and group‑based DRL (Yeku)
* Testing in more challenging settings (Z8su)
* Additional results on joint disentanglement & real‑world multi‑object dataset (Z8su,Yeku)

We clarify that:

* Our contribution is to disentangle factors via factor‑specific mixing strategies, without introducing factor‑specific architectural/objective biases. In contrast,
  * [1] requires an unscalable object-specific additive decoder, where our empirical results validate that [1] fails in attribute/object/joint disentanglement on complex scenes. We also clearly acknowledge adopting compositional consistency loss from [1] in revised paper.
  * Weakly supervised methods require extra information (e.g., paired observations with matched factors), often hard to obtain, whereas our main scope is unsupervised learning of disentangled factors.
  * Unlike group-based DRL using commutative properties on a single latent, we define group action on disentangled representation pairs, imposing necessary conditions for how factors combine to generate data, which cant’ be induced by single-latent group action.
* For more challenging scenarios, we verified that our method maintains disentanglement performance even with correlated factors.
* For additional experiments, on MSN dataset with 4 global styles, we confirm that our method achieves competitive object disentanglement and superior style disentanglement compared to baselines.

We appreciate the engagement of Z8su, 1m9R, and XmPJ, and are glad our responses resolved their concerns. Regarding Yeku’s concerns, while we have not received acknowledgement of our rebuttal from the reviewer, we believe our clarification on the comparison to group-based DRL and experiments on MSN dataset address the key concerns. We will add these experiments and clarifications to our paper. We thank the reviewers’ and AC’s constructive comments, which substantially improved our paper.

[1] Provable Compositional Generalization for Object-Centric Learning, ICLR24

---

### Decision · Program_Chairs · 2025-09-17

**Decision:**

Accept (poster)

**Comment:**

The paper makes a practical contribution to disentangled representation learning by introducing a modular compositional bias that supports attribute, object, and joint disentanglement under a unified framework. The approach is conceptually clear and empirically well-supported. While some concerns remain, including the lack of theoretical justification and overlap with prior work, these do not undermine the core practical contribution.